# Epi-microRNA mediated metabolic reprogramming counteracts hypoxia to preserve affinity maturation

Rinako Nakagawa [1] ✉, Miriam Llorian [2], Sunita Varsani-Brown[3],
Probir Chakravarty[2], Jeannie M. Camarillo[4], David Barry[5], Roger George[6],
Neil P. Blackledge[7], Graham Duddy[3], Neil L. Kelleher [4], Robert J. Klose [7],
Martin Turner [8] & Dinis P. Calado [1] ✉

To increase antibody affinity against pathogens, positively selected GC-B cells initiate cell division in the light zone (LZ) of germinal centers (GCs). Among these, higher-affinity clones migrate to the dark zone (DZ) and vigorously proliferate by utilizing energy provided by oxidative phosphorylation (OXPHOS). However, it remains unknown how positively selected GC-B cells adapt their metabolism for cell division in the glycolysis-dominant, cell cycle arrest-inducing, hypoxic LZ microenvironment. Here, we show that microRNA (miR)−155 mediates metabolic reprogramming during positive selection to protect high-affinity clones. Mechanistically, miR-155 regulates H3K36me2 levels in hypoxic conditions by directly repressing the histone lysine demethylase, *Kdm2a*, whose expression increases in response to hypoxia. The miR-155-*Kdm2a* interaction is crucial for enhancing OXPHOS through optimizing the expression of vital nuclear mitochondrial genes under hypoxia, thereby preventing excessive production of reactive oxygen species and subsequent apoptosis. Thus, miR-155-mediated epigenetic regulation promotes mitochondrial fitness in high-affinity GC-B cells, ensuring their expansion and consequently affinity maturation.

Germinal centers (GC) transiently form in B cell follicles of secondary lymphoid organs upon infection or vaccination. GCs are indispensable sites for increasing the affinity of antibodies, i.e., for the production of high-affinity antibody-secreting plasma cells that are critical for protecting hosts from pathogens. The process by which antibodies increase affinity to the antigen is called affinity maturation[1,2] and occurs in GCs through a bidirectional migration cycle between two regions, the light zone (LZ) and the dark zone (DZ)[3]. In the LZ, GC-B cells undergo antigen-driven selection in response to signals from B cell receptor (BCR) engagement and T follicular helper cells (TFH)[4]. Positively selected LZ GC-B cells induce *Myc*, a critical cell cycle regulator required for GC maintenance, and transient cMYC expression marks "licensed" GC-B cells[5,6]. Upon receiving positive selection signals, GC-B cells initiate cell division in the LZ. Among these cells, clones with higher-affinity further vigorously proliferate after their transition to the DZ for the clonal expansion[7]. Failure of the positive selection process disrupts GC responses, hinders affinity maturation, and consequently compromises humoral immunity.

[1]Immunity and Cancer Laboratory, Francis Crick Institute, London, UK. [2]Bioinformatics and Biostatistics Laboratory, Francis Crick Institute, London, UK. [3]Genetic Modification Service Laboratory, Francis Crick Institute, London, UK. [4]Department of Chemistry, Molecular Biosciences and the National Resource for Translational and Developmental Proteomics, Northwestern University, Evanston, IL, USA. [5]Advanced Light Microscopy Laboratory, Francis Crick Institute, London, UK. [6]Structural Biology Laboratory, Francis Crick Institute, London, UK. [7]Department of Biochemistry, University of Oxford, Oxford, UK. [8]Immunology Programme, The Babraham Institute, Cambridge, UK. ✉e-mail: rinako.nakagawa@crick.ac.uk; dinis.calado@crick.ac.uk

Mitochondrial oxidative phosphorylation (OXPHOS) is essential for GC-B cells to proliferate in the DZ[8,9]. In contrast, quiescent GC-B cells are relatively reliant on glycolysis[10]. This is concordant with recent reports showing that the LZ, where predominantly quiescent cells reside, represents a hypoxic microenvironment[11]. Hypoxia promotes glycolysis[12] and suppresses energy consuming processes, such as cell cycle progression and translation[13–15]. Therefore, cells under hypoxic conditions generally arrest the cell cycle. Nonetheless, engaged discussion surrounds the finding that only LZ GC-B cells reside in a hypoxic microenvironment. This is partly because conventional hypoxia probes are effective only when oxygen levels fall below 1%, and there is a heavy reliance on HIF-1 as a hypoxia marker. These limitations, along with model-specific variations, have led to conflicting reports[8,10,11,16,17], leaving uncertainty about whether positively selected GC-B cells experience hypoxia.

If the LZ is hypoxic, regardless of the hypoxic status in the DZ, positively selected GC-B cells must undergo metabolic reprogramming from quiescent to proliferative states by counteracting hypoxia. However, such mechanisms remain poorly understood. Moreover, the lack of clarity regarding the hypoxia status in GC-B cells further complicates the understanding of these mechanisms.

Here, we show that microRNA (miR)−155 functions as an epimiRNA, an epigenetic machinery regulating miRNA, that is essential for metabolic reprogramming in LZ GC-B cells during positive selection. By directly suppressing the hypoxia-induced lysine demethylase 2a (Kdm2a), which demethylates H3K36me2 to repress transcription[18,19], miR-155 promotes mitochondrial remodeling in high-affinity cMyc[+] GC-B cells under hypoxic conditions. The miR-155-Kdm2a interaction enhances OXPHOS by optimizing the expression of key nuclear mitochondrial genes, while preventing excessive production of reactive oxygen species (ROS) and subsequent apoptosis. Hence, miR-155 promotes mitochondrial fitness in high-affinity LZ GC-B cells during positive selection, which allows these cells to endure rapid proliferation upon transitioning to the DZ, thus ensuring affinity maturation.

## Results

### Mitochondrial ROS production is exacerbated by *Mir155*-deficiency in GC-B cells

OXPHOS is required to support the robust proliferation of DZ GC-B cells particularly with high affinity BCRs[9]. Mitochondrial ROS are produced during normal OXPHOS activity at the electron transport chain on the inner mitochondrial membrane[20]. However, defective mitochondria drive overproduction of ROS, which can activate apoptotic pathways[21]. MiR-155 is predominantly expressed in cMyc[+] GC-B cells and is known to protect cMyc[+] GC-B cells from apoptosis, particularly those with high-affinity BCRs[22]. Therefore, we hypothesized that miR-155 prevents cell death, in part, by limiting excessive production of mitochondrial ROS.

To test the hypothesis, we utilized the SW_HEL adoptive transfer system which allowed us to investigate miR-155 functions in GC-B cells in a B cell-intrinsic manner. In this system, B cells from *Mir155*-sufficient or *Mir155*-deficient SW_HEL *Myc*[gfp/gfp] *Aicda*-cre-hCD2 mice were transferred into CD45.1[+] congenic recipient mice. SW_HEL mice express the hen egg lysozyme (HEL)-specific immunoglobulin heavy chain knock-in and light chain transgene (HyHEL10 BCR)[23]. *Myc*[gfp/gfp] reporter mice express a GFP-cMYC fusion protein from the endogenous *Myc* locus[24]. *Aicda*-cre-hCD2 transgenic mice carry a Cre gene linked with a truncated human CD2 cDNA via an internal ribosome entry site (IRES) under the control of *Aicda* promoter[25]. In these mice, positively selected cMyc[+] GC-B cells were identified by GFP fluorescence and hCD2 surface expression. Recipient mice that received transferred B cells were challenged with HEL[3×] (Fig. 1a). Donor GC-B cells (CD45.1[−] CD45.2[+] B220[+] hCD2[+] CD38[lo] CD95[+]) were gated as described in Supplementary Fig. 1a, and by coupling DZ/LZ distinguishing markers (CXCR4/CD86) to cMyc-GFP expression. Then, GC-B cells were

subdivided into four consecutive subpopulations: cMyc[−] LZ, cMyc[+] LZ, cMyc[−] DZ and cMyc[+] DZ[7]. MiR-155 is mostly co-expressed with cMYC in GC-B cells, and the expression of both cMYC and miR-155 changes as follows: highest levels in cMyc[+] LZ GC-B cells, intermediate levels in cMyc[+] DZ GC-B cells and followed by negligible levels in cMyc[−] DZ and cMyc[−] LZ GC-B cells[5,6,22] (Supplementary Fig. 1b). Lack of miR-155 did not impact the expression of cMYC in GC-B cells, hence the mean fluorescent intensity (MFI) of cMyc-GFP was comparable between wildtype (hereafter called as "WT") and *Mir155*-knock out (hereafter called as "KO") cells (Supplementary Fig. 1c). However, the number of cMyc[+] LZ and cMyc[+] DZ GC-B cells were significantly reduced in the absence of miR-155 (Supplementary Fig. 1d). The MitoSOX-based flow cytometric assay for the detection of mitochondrial ROS revealed that in the absence of miR-155, GC-B cell subpopulations, including cMyc[+] LZ, cMyc[+] DZ and cMyc[−] DZ, exhibited a significantly higher percentage of MitoSOX[+] cells compared to their respective WT cells (Fig. 1b, Supplementary Fig. 1e). Additionally, we observed a concurrent increase in the proportion of active-caspase 3[+] cells alongside decreased proliferation within the cMyc[+] LZ, cMyc[+] DZ and cMyc[−] DZ subpopulations of the KO mice compared to the WT mice (Fig. 1c, Supplementary Fig. 1f). These findings indicate that miR-155 plays a role in mitigating mitochondrial ROS levels in GC-B cells.

### MiR-155 is required for promoting mitochondrial fitness in GC-B cells

To gain a comprehensive understanding of miR-155-mediated regulation of mitochondrial functions and dynamics of OXPHOS activity during the migration cycle of GC-B cells, we conducted single cell (sc) RNA-seq on both WT and KO GC-B cells and dissected GC-B cells into distinct subpopulations. Unsupervised clustering identified nine distinct clusters: LZ signature genes[9] were expressed in clusters 0, 1, 2, 5 and 8, and clusters 0 and 1 predominantly represented cMyc[−] LZ GC-B cells due to their enrichment in the G1 phase of the cell cycle (Supplementary Fig. 1g, h, i). Cluster 8 showed enrichment in pre-memory B cell (pre-MBC)-associated genes, such as *Ccr6*, *Cd38* and *Hhex*[26,27]), indicating its representation of pre-MBC cells (Supplementary Data 1). DZ signature genes[9] were expressed in clusters 3, 4, 5, 6 and 7 (Supplementary Fig. 1i), which correlated strongly with cell cycle progression, i.e., cluster 6; S phase, cluster 4; S→G2/M phase, cluster 7; G2/M phase and cluster 3; G2/M→G1 phase (Supplementary Fig. 1g). The DZ clusters, 3, 4, 6 and 7 were further analyzed by RNA velocity[28] to determine a temporal sequence of DZ GC-B cells. This analysis shows the order of the DZ GC-B cell clusters as clusters 6 → 4 → 7 → 3 (Supplementary Fig. 1g).

To establish more specific relationships between the clusters we employed partition-based graph abstraction (PAGA)[29]. The PAGA graph revealed that LZ clusters 0, 1, 2 and 8 formed a closely interconnected group, while DZ clusters 3, 4, 6 and 7, characterized by high expression of Cxcr4, formed a distinct group (Fig. 1d and Supplementary Fig. 1j). Cluster 5, which exhibited higher mitochondrial content, but lower gene counts than the other clusters, did not belong to either LZ or DZ groups (Fig. 1d, Supplementary Fig. 1j, k). Notably, this cluster displayed a significant decrease in rRNA levels (Supplementary Fig. 1l), which is indicative of early apoptosis[30]. Therefore, we excluded cluster 5 from further analysis. The high expression of *Myc* in clusters 2 (LZ) and 6 (DZ) in the PAGA graph corresponded to the enrichment of *Myc* signature genes[5,6] observed in these clusters, as shown in the dot plot (Fig. 1d and Supplementary Fig. 1i). The increased *Myc* expression in cluster 8 (pre-MBCs) in Fig. 1d is consistent with our previous work suggesting that *Myc*-induced positively selected GC-B cells can give rise to MBCs[7,31]. In the PAGA graph, clusters 0 (LZ) and 3 (DZ) predominantly connect the LZ and DZ groups (Fig. 1d and Supplementary Fig. 1j), indicating that cluster 0 represents LZ B cells soon after migration back from the DZ. The temporal and spatial dynamics of the clusters are summarized in Fig. 1e.

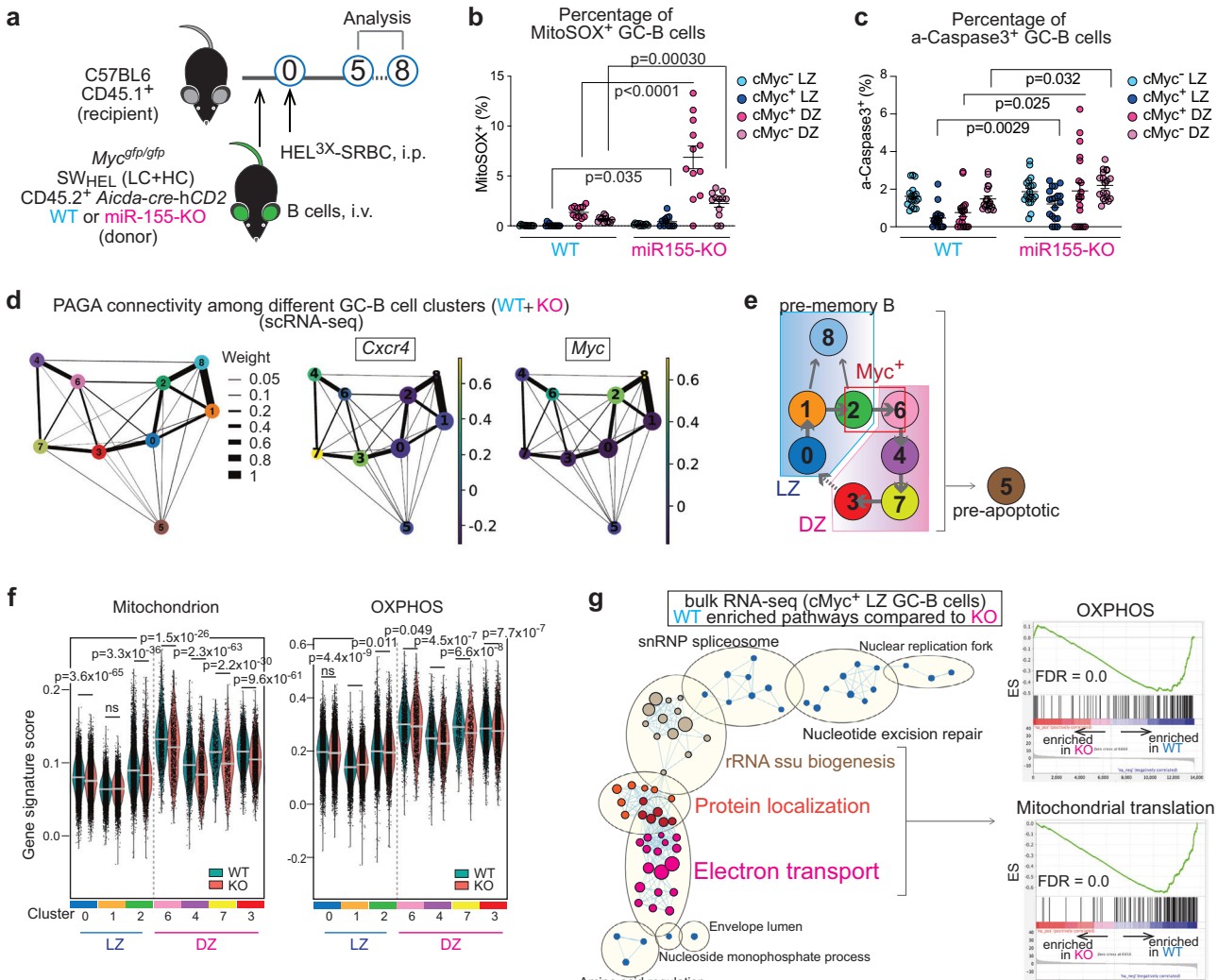

**Fig. 1 | MiR-155 is required for promoting mitochondrial fitness in GC-B cells.**
**a** Experimental design for all the experiments for Fig. 1. **b** Percentage of MitoSOX[+] GC-B cells five days after HEL[3×]-SRBC immunization. Pooled from three independent experiments (WT $n = 12$; KO $n = 12$). Unpaired student's t-test, two-tailed. **c** Percentage of active-Caspase 3[+] GC-B cells at seven days after HEL[3×]-SRBC immunization. Pooled from two independent experiments (WT $n = 18$; KO $n = 19$). Unpaired student's t-test, two-tailed. **d** The PAGA algorithm generates a topology-preserving map of single cells along with the sequential changes of gene expression. PAGA graphs were constructed using a k-nearest neighbors' algorithm to show relationships between clusters. The Ball-and-stick representation displays PAGA connectivity among GC-B cell clusters identified though scRNA-seq. Ball size represents cluster size, and edge thickness indicates connectivity between clusters. Normalized gene expression for *Cxcr4* or *Myc* is shown in the PAGA graph. **e** Predicted relationship between the clusters identified in scRNA-seq of GC-B cells. **f** Violin plots showing gene signature scores for "Mitochondrion" and "OXPHOS". The bar represents median value. Wilcoxon rank sum test, two-sided, n.s., not significant. **g**, Enrichment map visualizing GSEA results of the bulk RNA-seq data from WT and KO cMyc[+] LZ GC-B cells. GC-B cells were sorted five days after HEL[3×]-SRBC immunization. WT-enriched pathways (q < 0.0001) were used. Nodes represent gene sets, and edges represent mutual overlap. Highly redundant gene sets were grouped as clusters (left). ES; enriched score, FDR; false discovery rate. Unless otherwise stated, mean ± SEM is indicated.

Using these temporally linked GC-B cell clusters, we investigated the changes in mitochondrial functions in GC-B cells in the absence of miR-155 by calculating the gene signature scores for "mitochondrion" and "OXPHOS". We found these to be significantly lower in cluster 2 (cluster enriched with cMyc[+] LZ GC-B cells) and all the DZ clusters (clusters 6, 4, 7 and 3) of KO cells than those of WT cells (Fig. 1f). These results suggest that the absence of miR-155 led to impaired mitochondrial functions in cMyc[+] LZ GC-B cells (cluster 2) and cMyc[+] DZ GC-B cells (cluster 6), and that such impairment was still observed in cMyc[-] DZ GC-B cells (clusters 4, 7 and 3) that no longer express miR-155 in WT mice. Thus, miR-155 plays a critical role in maintaining mitochondrial integrity in cMyc[+] GC-B cells and likely impacts the bioenergetic functions in GC-B cells.

To further clarify the regulatory role of miR-155 in LZ GC-B cells, we carried out bulk RNA-seq analysis on WT and KO cMyc[+] LZ GC-B cells. Gene set enrichment analysis (GSEA) on RNA-seq data revealed the top eight enriched biological pathways in WT cMyc[+] LZ GC-B cells compared to KO cMyc[+] LZ GC-B cells were related to "ribosome", "translation" and "mitochondria", either alone or in combination (Supplementary Fig. 1m). Nearly half of the significantly increased genes in the top three pathways ("ribosomal subunit", "Structural constituent of ribosome" and "ribosome") encoded mitochondrial ribosomal proteins (Supplementary Data 2). Mitochondrial ribosomal proteins are encoded by nuclear genes and are subsequently transported to the mitochondria following their synthesis in the cytoplasm[32]. The Cytoscape network analysis of GSEA results revealed an enrichment of pathways involved in mitochondrial protein transport and mitochondrial functions, particularly electron transport, in WT cMyc[+] LZ GC-B cells compared to their KO counterparts (Fig. 1g). This finding indicates that these pathways are compromised or

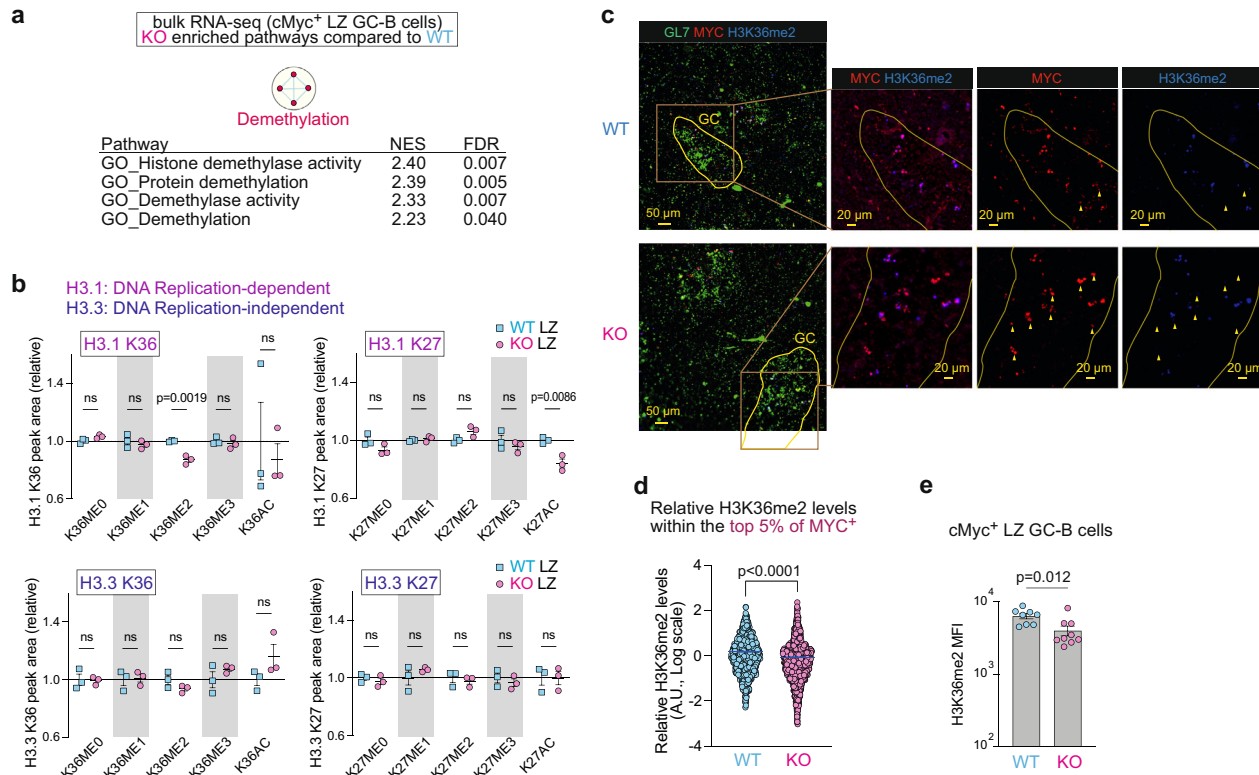

**Fig. 2 | MiR-155 regulates H3K36me2 levels in dividing LZ GC-B cells.**
**a** Enrichment map visualizing GSEA results of bulk RNA-seq data from WT and KO cMyc⁺ LZ GC-B cells. KO-enriched pathways (q < 0.1) were used. **b** The relative abundance of histone PTMs, including histone H3.1 K36, H3.3 K36, H3.1 K27 and H3.3 K27 in LZ GC-B cells from WT and KO mice. Donor-derived GC-B cells were sorted five days after HEL³ˣ-SRBC immunization for analysis. Triplicate samples were used for both WT and KO backgrounds. Unpaired student's t-test, two-tailed. **c** Splenic sections of C57BL/6 mice, in the presence and absence of miR-155, seven days after SRBC immunization were stained for GL7 (green), MYC (red) and H3K36me2 (blue). Yellow wedges represent cells that are positive for MYC but

negative for KDM2A. Scale bar, 50 μm. Representative images from two experiments (WT n = 4; KO n = 4). **d** Relative H3K36me2 levels determined by calculating the ratio of H3K36me2 MI to MYC MI from Supplementary Fig. 2c. The top 5% of MYC intensity values, representing the highest MYC expressing cells, were selected from each batch of two independent experiments. The bar represents mean. Unpaired student's t-test, two-tailed. A.U., arbitrary unit. **e** MFI of H3K36me2 in cMyc⁺ LZ GC-B cells. Pooled from three independent experiments (WT n = 8; KO n = 9). Unpaired student's t-test, two-tailed. Unless otherwise stated, mean ± SEM is indicated. n.s., not significant.

defective in KO cMyc⁺ LZ GC-B cells. In line with the enrichment map, genes associated with "OXPHOS", and "Mitochondrial translation" exhibited a significant enrichment in WT cMyc⁺ LZ GC-B cells compared to KO cMyc⁺ LZ GC-B cells, suggesting impaired functionality of these pathways in KO cMyc⁺ LZ GC-B cells (Fig. 1g). To assess metabolism in GC-B cells, we conducted SCENITH, which allows metabolic profiling by flow cytometry[33]. The results showed that KO cMyc⁺ GC-B cells exhibited increased glucose dependence (i.e., reliance on glycolysis and OXPHOS) and reduced fatty acid and amino acid oxidation (FAO/AAO) compared to their WT counterparts (Supplementary Fig. 1n). This metabolic profile is consistent with that observed in GC-B cells from transcription factor A, mitochondrial (*Tfam*)-deficient mice, which have defective mitochondria[34]. These findings strongly indicate impaired mitochondria in KO cMyc⁺ GC-B cells. Collectively, our findings suggest that mitochondrial fitness is exclusively established in the LZ during positive selection and that miR-155 expressed in the LZ is critical for maintaining mitochondrial bioenergetic functions in GC-B cells.

## MiR-155 regulates H3K36me2 levels in dividing LZ GC-B cells

To visualize the enriched pathways in KO cells, we generated a Cytoscape enrichment map using RNA-seq data from cMyc⁺ LZ GC-B cells. The enrichment map revealed that the gene signature "Demethylation" was the only pathway significantly enriched in KO cMyc⁺ LZ GC-B cells when compared to WT cells (Fig. 2a). The signature

"Demethylation" represents the process of removing one or more methyl groups from a molecule and consisted of four pathways in this comparison: "Histone demethylase activity", "Protein demethylation", "Demethylase activity", and "Demethylation" (Fig. 2a, Supplementary Data 3). Therefore, we investigated whether there were any alterations in histone post-translational modifications (PTM) in KO GC-B cells compared to WT GC-B cells. To this end, we employed a liquid chromatography with tandem mass spectrometry (LC-MS/MS) technique that allows unbiased identification and quantitative profiling of histone PTMs[35,36]. This method successfully quantified the abundance of histone PTMs in both WT and KO LZ GC-B cells, including methylation and acetylation on histone H1.4, H3 and H4. The analysis showed a significant decrease in di-methylation of histone H3.1 at lysine 36 (K36me2) and an increase in unmethylated histone H3 at lysine 23 (K23me0) in KO LZ GC-B cells compared to WT LZ GC-B cells (Fig. 2b and Supplementary Fig. 2a). We also observed a decrease in acetylation of H3 at K23 (K23ac) (Supplementary Fig. 2a), which is linked to transcriptional activation[37]. Acetylation of H3 at K23 can be catalyzed by cyclic adenosine monophosphate response element-binding protein binding protein (CREBBP), monocytic leukemia zinc finger protein (MOZ) complex and its paralog monocytic leukemia zinc finger related factor (MORF) complex[37,38]. However, there were no differentially expressed genes (DEG) in the bulk RNA-seq dataset between KO and WT cMyc⁺ LZ-GC-B cells for any genes encoding protein products that are part of these complexes (Supplementary Data 2). Therefore, it was

unlikely that the absence of miR-155 was directly responsible for the decreased H3K23ac levels in KO LZ GC-B cells. Because the reduction of H3K23ac could lead to increased levels of unmethylated H3K23, the elevation in H3K23me0 levels appears to be not directly caused by the lack of miR-155 either. Thus, the decreased levels of K36me2 on histone H3.1 appeared to be the sole altered histone methylation directly impacted by the absence of miR-155-mediated regulation in KO cMyc$^+$ LZ GC-B cells.

The reduction of K36me2 levels was specific to DNA replication-dependent histone H3.1 and was not observed in DNA replication-independent histone H3.3 (Fig. 2b). This suggests that cell division is required for the depletion of H3K36me2. Previous studies have indicated that the loss of H3K36me2 is associated with a decrease in acetylation of histone H3 at K27 (K27ac)[39]. There was decreased K27ac on histone H3.1 in KO LZ GC-B cells relative to WT cells, while no such change was observed on histone H3.3 (Fig. 2b). The decrease in K36me2 on histone H3.1 may lead to increased levels of mono-methylated K36 (K36me1) on histone H3.1. While no elevation of K36me1 was detected on histone H3.1 in KO LZ GC-B cells relative to their WT cells, a significant increase in this histone PTM was observed in KO DZ GC-B cells compared to their WT counterparts (Fig. 2b and Supplementary Fig. 2b). This could be attributed to the migration of dividing LZ GC-B cells to the DZ, where chromatin marks could potentially be deposited on newly synthesized histones. Apart from K36me1, no other changes were observed in the histone PTMs at K36 of histone H3.1 in DZ GC-B cells between WT and KO cells (Supplementary Fig. 2b). These findings suggest that miR-155 regulates the levels of K36me2 of H3.1 during cell division in LZ GC-B cells. Given that LZ GC-B cells are generally quiescent, these dividing cells likely represent cMyc$^+$ LZ GC-B cells[7].

To examine the effect of miR-155-mediated regulation on histone H3K36me2 levels in cMyc$^+$ GC-B cells, we performed immunostaining of spleen sections from SRBC-immunized, Mir155-sufficient and Mir155-deficient mice. Within the GL7-positive GC areas, a correlation in co-expression of MYC and H3K36me2 was observed in both groups to some extents (Fig. 2c, Supplementary Fig. 2c). However, the Pearson correlation coefficients with 95% confidence interval showed no overlap between the values of WT and KO groups (r = 0.574 − 0.596 and 0.344 − 0.384, for WT and KO, respectively), indicating a stronger correlation in the WT group than the KO group (Supplementary Fig. 2c). To investigate the difference in H3K36me2 levels between WT and KO MYC$^+$ GC-B cells, we compared the relative mean intensity of H3K36me2 in cells within the top 5% of MYC positive cells from the total cell population shown in Supplementary Fig. 2c. The analysis revealed a significant reduction in H3K36me2 levels in KO MYC$^+$ GC-B cells compared to the corresponding WT cells (Fig. 2d). This reduction of H3K36me2 levels in KO cMyc$^+$ LZ GC-B cells was also confirmed by flow cytometry (Fig. 2e). Consistent with the results derived from the MS analysis (Fig. 2b), total LZ GC-B cells in KO background showed reduced H3K36me2 levels compared to WT cohorts, whereas total DZ GC-B cells did not show this reduction (Supplementary Fig. 2d). We also performed immunofluorescence-based correlation analysis between MYC and H3K36me1 or H3K36me3. However, the correlation in co-expression of MYC and these histone PTMs exhibited only minor variations when comparing KO and WT MYC$^+$ GC-B cells (Supplementary Fig. 2e, f). These results suggest that the co-expression pattern between MYC and H3K36me1 or H3K36me3 remains largely unchanged, unlike the substantial changes observed in H3K36me2. Collectively, these results provide compelling evidence for the crucial role of miR-155 as an epigenetic modulator in histone PTMs. Notably, miR-155 predominantly regulates the levels of H3K36me2 in dividing cMyc$^+$ LZ GC-B cells, potentially this may influence mitochondrial functions in GC-B cells.

## MiR-155 primarily controls Kdm2a expression in high-affinity cMyc$^+$ LZ GC-B cells

Demethylation of H3K36me2 can be catalyzed by the lysine demethylases (KDMs), Kdm2a, Kdm2b, Kdm4a, Kdm4b and Kdm4c[40]. We examined whether any of mRNAs encoding these enzymes was differentially expressed between WT and KO cMyc$^+$ LZ GC-B cells. Expression of Kdm2a was significantly higher in KO cMyc$^+$ LZ GC-B cells compared to WT cMyc$^+$ LZ GC-B cells, while the others (Kdm2b, Kdm4b and Kdm4c) exhibited similar expression levels between the two groups (Fig. 3a). In contrast, the expression of Kdm4a was found to be decreased in KO cMyc$^+$ LZ GC-B cells compared to WT cMyc$^+$ LZ GC-B cells (Fig. 3a and Supplementary Data 2). Next, we examined whether Kdm2a mRNA is predicted as a target of miR-155. By utilizing miR prediction tools, we identified Kdm2a as one of the 18 genes that were predicted to be targets of miR-155 and showed differential expression between WT and KO cMyc$^+$ LZ GC-B cells (Supplementary Fig. 3a and Supplementary Data 2). Kdm2a has one 8-mer miR-155-binding site at 3' UTR that is conserved across multiple species, including human, mouse, and rat (Fig. 3b).

To investigate the relationship between KDM2A and miR-155, we examined the co-expression of KDM2A and MYC by immunostaining tissue sections derived from Mir155-sufficient and Mir155-deficient mice using antibodies against IgD, MYC and KDM2A. This revealed a strong correlation in co-expression of these two proteins in both groups (r = 0.775−0.788 and 0.720−0.740, for WT and KO, respectively) (Fig. 3c, d). Therefore, we concluded that KDM2A is mainly expressed in positively selected cMyc$^+$ GC-B cells. To examine the degree and cell subpopulations expressing KDM2A, we performed flow cytometric analysis of GC-B cells derived from SW$_{HEL}$ adoptive transfer system. In WT mice, the analysis revealed a ~1.3-fold increase in KDM2A expression in cMyc$^+$ LZ GC-B cells compared to cMyc$^-$ LZ GC-B cells (Fig. 3e), indicating that positive selection signals can induce its expression. The highest expression of KDM2A was observed in cMyc$^+$ DZ GC-B cells. A similar expression pattern was observed in KO GC-B cell subpopulations. In all GC-B cell subpopulations, KDM2A expression was significantly higher in KO cells compared to their WT counterparts (Fig. 3e and Supplementary Fig. 3b).

The SW$_{HEL}$ system enables the distinction between high and low affinity B cells following HEL$^{3x}$ immunization. When comparing high and low affinity clones within each WT GC-B cell subpopulation, there were no significant differences in the level of KDM2A (Fig. 3f). In contrast, the lack of miR-155 led to a substantial increase of KDM2A expression in high affinity clones compared to low affinity clones within the cMyc$^+$ LZ GC-B cell subpopulation. As a result, there was a marked elevation in KDM2A expression in high affinity clones of KO cells compared to WT cells (Fig. 3f). Collectively, these results suggest that KDM2A is preferentially induced in high affinity clones within the cMyc$^+$ LZ GC-B cell subpopulation during positive selection, and therefore, miR-155 is required to dampen KDM2A expression in these cells.

## Hypoxia induces KDM2A in LZ GC-B cells

Hypoxia induces Kdm2a expression at both transcript and protein levels[41,42]. Given that previous studies have shown that the LZ of GCs is hypoxic[11], we hypothesized that miR-155 counteracts the hypoxia-induced increase of KDM2A expression: i.e., in the absence of miR-155, KDM2A expression is enhanced under hypoxic conditions. Since the concept that the LZ has a more hypoxic microenvironment than the DZ remains a subject of controversy[8,16], we quantified the degree of hypoxia that GC-B cells were experiencing using the hypoxia detecting dye, MAR (see "methods"), which effectively differentiated between hypoxic and normoxic B cells (Supplementary Fig. 4a). Flow cytometric analysis revealed that a bimodal distribution of MAR fluorescence intensity in non-GC-B cells. This potentially reflects their

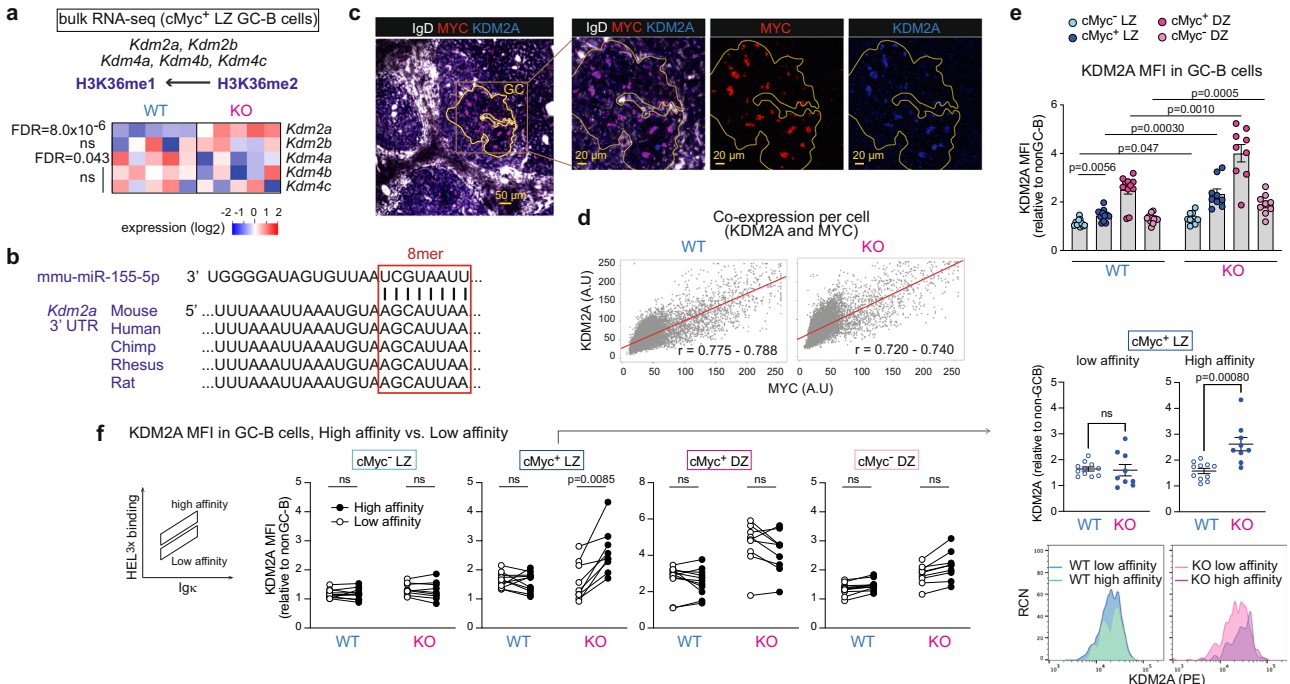

**Fig. 3 | MiR-155 primarily controls KDM2A expression in high affinity cMyc⁺ LZ GC-B cells. a** Heat map showing the transcript levels of the enzymes involved in the demethylation of histone H3K36me2. The bulk RNA-seq dataset from WT and KO cMyc⁺ LZ GC-B cells was used. FDR values are shown. **b** Sequence alignment of a part of the 3'UTR of *Kdm2a*. The 8-mer miR-155 binding site is highlighted in a box. **c** Immunofluorescence staining of splenic section from a WT mouse seven days after SRBC immunization, showing IgD (white), MYC (red) and KDM2A (blue). Scale bar, 50 μm. Representative section images from two experiments (WT $n = 4$; KO $n = 4$). **d** Scatter plots show the MI of MYC and KDM2A within segmented GC cells in spleen sections from immunized WT and KO mice. The Pearson correlation

coefficient (r) is presented with a 95% confidence interval in red. A.U., arbitrary unit. **e** Relative MFI of KDM2A in cells from four GC-B cell subpopulations. GC-B cells were derived from WT or KO donor B cells seven days after HEL³ˣ-SRBC immunization. Pooled from two independent experiments (WT $n = 11$; KO $n = 9$). Unpaired student's t-test, two-tailed. **f** Relative MFI of KDM2A in cells from four GC-B cell subpopulations. GC-B cells were derived from WT or KO donor B cells seven days after HEL³ˣ-SRBC immunization. Low and high affinity were determined based on HEL³ˣ binding within Igκ⁺ GC-B cells. Pooled from two independent experiments (WT $n = 11$; KO $n = 9$). RCN, relative cell number. Paired student's t-test, two-tailed. Unless otherwise stated, mean ± SEM is indicated. n.s., not significant.

location within the tissue, as regions adjacent to blood vessels are generally considered less hypoxic, whereas areas distant from blood vessels experience more hypoxia[43]. Additionally, differences in detection sensitivity between MAR (which detects hypoxia at ~5% $O_2$)[44] and conventional hypoxia probes, such as pimonidazole (which detects hypoxia at <1% $O_2$)[45], may contribute to the detection of non-GC-B cells as hypoxic. Here, LZ GC-B cells exhibited higher MAR fluorescence intensity than DZ GC-B cells (Fig. 4a), indicating greater levels of hypoxia, in line with previous studies[10,11,17].

To examine the effect of hypoxia on KDM2A expression in B cells, we cultured splenic B cells in the presence of anti-IgM, anti-CD40 and IL-4, which mimicked the combination of positive selection signals derived from BCR and T cell help. These cells were cultured in either normal oxygen concentration (21%, normoxic) or low oxygen concentration (1%, hypoxic), and KDM2A expression was examined by flow cytometry. Following a two-day activation with positive selection mimics under normoxic conditions, both WT and KO B cells showed increased KDM2A expression (Fig. 4b). However, there was no significant difference in KDM2A levels between WT and KO B cells under normoxic conditions (Fig. 4b). Under hypoxic conditions, the expression of KDM2A was markedly increased, resulting in higher expression of KDM2A in both WT and KO B cells compared to normoxic conditions (Fig. 4b). By contrast, in hypoxic cultures, KDM2A expression was ~1.6-fold greater in the KO B cells than WT B cells (Fig. 4b). This difference in expression was attributed to the increased expression of miR-155 under hypoxic conditions (Supplementary Fig. 4b), consistent with previous reports[46,47]. Therefore, hypoxia was necessary for observing differential levels of KDM2A between KO and WT in vitro

cultured B cells. These findings suggest that miR-155 exerts a strong repressive effect on *Kdm2a* primarily under hypoxic conditions.

## The regulatory activity of miR-155 is revealed under hypoxia

To examine the impact of hypoxia on cell division, we cultured Cell-TraceViolet (CTV)-labelled B cells from *Mir155*-sufficient and *Mir155*-deficient mice with positive selection mimics in either normoxic or hypoxic conditions. As expected, hypoxia suppressed cell proliferation regardless of genotype (Fig. 4c). However, whereas both WT and KO B cells showed similar levels of division in normoxic cultures, miR-155 KO B cells exhibited significantly less division than WT B cells under hypoxic conditions (Fig. 4c). In contrast to ex vivo GC-B cells, the levels of apoptosis in WT cells and KO B cells were similar in vitro (Supplementary Fig. 4c, Fig. 1c). The results highlight the critical role of miR-155 for cell proliferation under hypoxic conditions.

To investigate the influence of hypoxia on mitochondrial ROS production, we used non-fluorescent MitoTracker Orange CM-$H_2$ TMRos that becomes fluorescent upon oxidization. After a two-day culture period under either hypoxic or normoxic conditions, we stained WT and KO B cells with MitoTracker dyes. WT B cells produced lower amounts of mitochondrial ROS under hypoxic conditions (Fig. 4d, e). In contrast, hypoxic KO cultures had significantly higher amounts of mitochondrial ROS than normoxic KO cultures (Fig. 4d, e), indicating mitochondrial dysfunction in these cells. When we compared the size of WT and KO B cells cultured under normoxia and hypoxia by flow cytometry, both WT and KO cells showed equally reduced cell size under hypoxia compared to normoxia (Supplementary Fig. 4d). Despite the reduced size of hypoxic cultures compared to

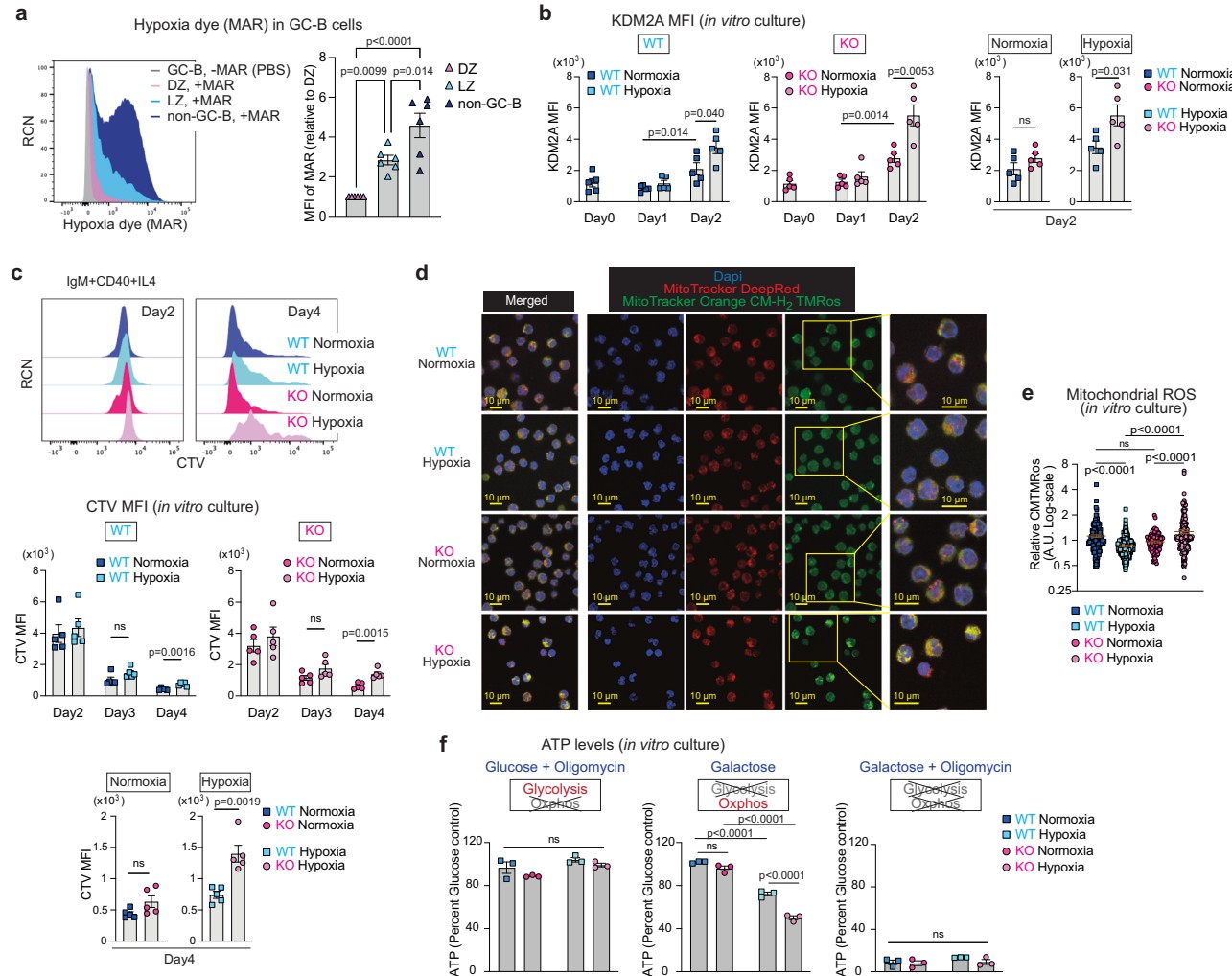

**Fig. 4 | The regulatory activity of miR-155 is revealed under hypoxia.**
**a** Representative flow cytometric histograms of hypoxia dye, MAR vs. relative cell number (RCN) (left). Relative MFI of MAR (right) in GC-B and non-GC-B cells. GC-B and non-GC-B cells were isolated from inguinal lymph nodes of C57BL/6 mice. One-way ANOVA. Pooled from three independent experiments ($n = 6$). **b** KDM2A MFI of B cells from WT and KO mice cultured under normoxia (21% $O_2$) or hypoxia (1% $O_2$) for the indicated time periods. Unpaired student's t-test, two-tailed. Pooled from three independent experiments (WT $n = 5$; KO $n = 5$). **c** Representative flow cytometric histograms of CTV vs. RCN at day 2 and 4 (top). CTV MFI of B cells from WT and KO mice cultured under normoxia or hypoxia for the indicated time periods (bottom). Unpaired student's t-test, two-tailed. Pooled from five independent experiments (WT $n = 5$; KO $n = 5$). **d** Representative image of B cells from WT and KO

mice cultured under normoxia or hypoxia for two days, followed by staining with MitoTracker Deep Red (Red) and MitoTracker Orange CM-H$_2$ TMRos (green). DAPI (blue) was used as a nuclear counterstaining. Scale bar, 10 μm. Representative images are shown. **e** Quantification of relative MitoTracker Orange CMTMRos levels. Each dot represents one cell. One-way ANOVA. Pooled from two independent experiments (WT $n = 4$; KO $n = 4$). **f** Relative ATP levels in B cells from WT and KO mice. The replated cells were cultured for two hours in hypoxia with indicated media for ATP measurement. The luminescence values of each type of B cells cultured in media containing glucose for two hours in hypoxia were used to calculate relative ATP levels. One-way ANOVA. Pooled data from three experiments (WT $n = 3$; KO $n = 3$). Unless otherwise stated, mean ± SEM is indicated. n.s., not significant.

normoxic cultures, we also observed increased ROS in hypoxic KO cultures compared to normoxic KO cultures through a MitoSOX-based flow cytometric assay (Supplementary Fig. 4e). Consequently, KO B cells exhibited a significantly increased ROS production compared to WT B cells under hypoxic conditions, while no such difference was observed under normoxic conditions (Fig. 4d, e, Supplementary Fig. 4e). To further investigate mitochondrial integrity in KO B cells, we assessed energy metabolic states coupled to mitochondrial functions. To this end, we measured the abundance of adenosine triphosphate (ATP) produced through metabolic processes. After two days of culturing WT and KO B cells under either normoxic or hypoxic conditions, we further cultured the cells in the following media supplemented with either glucose, glucose plus oligomycin, galactose, or galactose plus oligomycin. To directly compare energy metabolic capacities of cells, all cells were cultured in hypoxic conditions for 2 h using the indicated

media. Cells can use both glycolysis and OXPHOS to produce ATP when cultured in glucose-supplemented medium, while in medium supplemented with glucose and oligomycin, an inhibitor of OXPHOS, cells rely solely on glycolysis for ATP production. Under hypoxic conditions, both WT and KO cells exhibited a reduction of ATP abundance when cultured in glucose-supplemented medium, compared to their normoxic counterparts (Supplementary Fig. 4f). This result is consistent with previous reports showing reduced energy production under hypoxia[48,49]. When B cells were cultured in medium supplemented with both glucose and oligomycin, the relative abundance of ATP was similar to corresponding cultures supplemented with glucose alone across all cell types (Fig. 4f), indicating that the cells were able to survive by relying on their glycolytic capacity without the need for OXPHOS, in accordance with the well-known "Crabtree effect"[50,51]. In contrast, cells cultured in galactose-supplemented medium must use

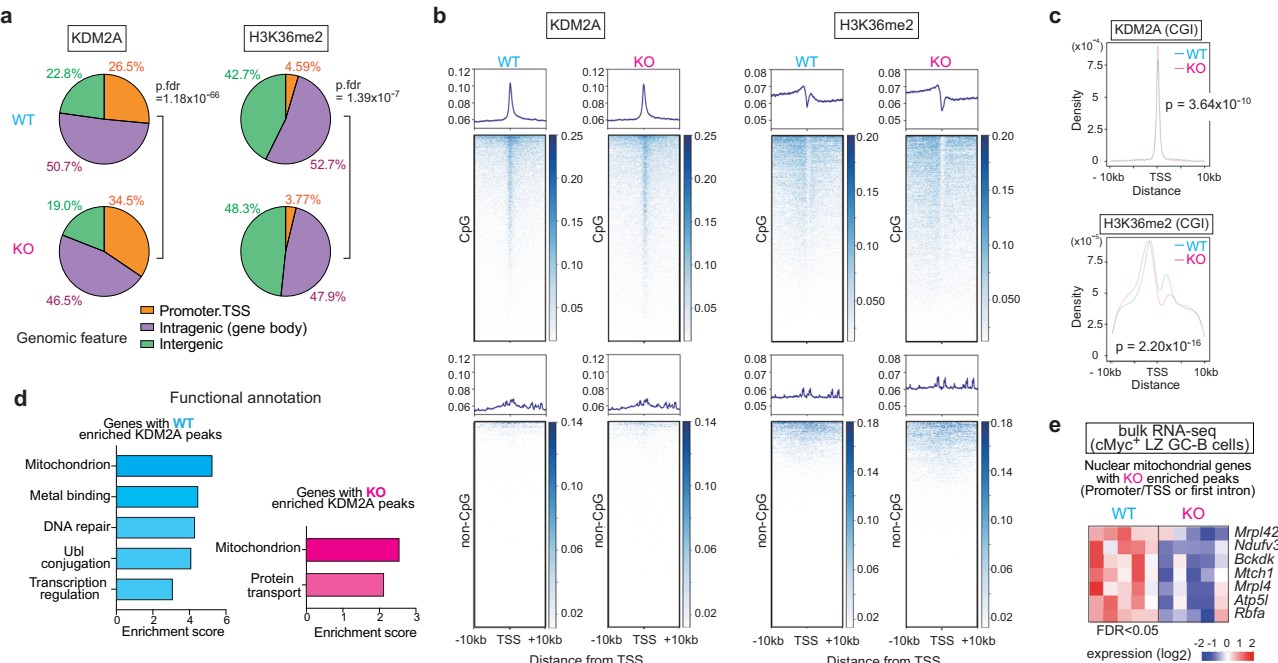

**Fig. 5 | MiR-155 optimizes nuclear mitochondrial gene expression through *Kdm2a*-mediated control of H3K36me2 levels. a** Genomic distribution of KDM2A and H3K36me2 ChIP-seq peaks in B cells from WT and KO mice cultured for two days under hypoxia is shown as pie charts. The average percentage of the ChIP-seq triplicate samples is displayed. Promoter regions are defined as ± 2 kb of the transcription start site (TSS). Two-sided, p.FDR values are shown. **b** Metagene analysis and heatmaps of KDM2A or H3K36me2 for CpG islands (CGIs) containing genes or genes without CGIs. Metagene analyses are displayed in ± *10 kb* windows.

A representative sample is shown. **c** Plots showing the distribution of peak distance to the nearest TSS. The average ChIP-seq read occupancy for KDM2A or H3K36me2 surrounding the TSS of CGIs containing genes. Kolmogorov-Smirnov test. *p*-values are shown. **d** Analysis of genes with "WT-enriched" and "KO-enriched" KDM2A peaks using the DAVID Functional Annotation Clustering Tool. Pathways with a two-sided modified fisher exact *p*-value < 0.05 were used for the analysis. **e** A heat map showing transcript levels of seven mitochondrial genes with "KO-enriched" KDM2A peaks at the promoter/TSS regions or first introns near the TSS.

OXPHOS to generate ATP[52]. In this culture condition, we observed a reduction in ATP abundance in both WT and KO B cells under hypoxia compared to normoxia.

Furthermore, while ATP production through OXPHOS was compromised in hypoxic cultures of both WT and KO cells, the impairment was more severe in KO B cells (Fig. 4f). ATP production was abrogated when WT or KO B cells were cultured in medium containing galactose and oligomycin regardless of oxygen tension (Fig. 4f). These results suggest that miR-155 plays a critical role in the metabolic reprogramming of activated B cells by regulating OXPHOS under hypoxic conditions. Collectively, the results demonstrate that the regulatory activity of miR-155 in modulating mitochondrial function and energy metabolism in activated B cells is manifested in a hypoxic environment.

## MiR-155 optimizes nuclear mitochondrial gene expression through KDM2A-mediated control of H3K36me2 levels

To investigate how *Mir155*-deficiency impacts genome-wide binding of KDM2A and H3K36me2 distribution, we performed ChIP-seq assays using hypoxic B cell cultures. The genomic distribution of KDM2A and H3K36me2 peaks showed significant differences between WT and KO B cells (Fig. 5a). In KO B cells compared to WT B cells, there was a proportional increase in KDM2A peaks within the promoter/transcriptional start site (TSS) regions (WT = 26.5%, KO = 34.5%), accompanied by a concurrent reduction of H3K36me2 peaks in the same regions (WT = 4.59%, KO = 3.77%) (Fig. 5a). Consistent with a previous report showing that KDM2A preferentially binds to CGIs islands (CGIs) primarily located at gene promoters[53], we detected significant enrichment of KDM2A peaks specifically at the TSS-proximal regions of genes containing CGIs (Fig. 5b). In agreement with these results, when the levels of H3K36me2 within 2.5 kb downstream of a TSS of

genes containing CGIs were calculated and presented as a ratio of average KO to WT for each gene, KO B cells exhibited lower H3K36me2 levels than WT B cells in the TSS regions of 17,618 genes. In contrast, KO B cells exhibited higher H3K36me2 levels in the TSS regions of only 6,724 genes (Supplementary Data 4). Furthermore, we observed a concurrent depletion in H3K36me2 signals within these regions of CGI containing genes in both WT and KO B cells (Fig. 5b). To identify distinctive patterns of KDM2A binding and H3K36me2 marks, we examined the peak positions in relation to the nearest TSS (Fig. 5c). These plots revealed that a significantly higher density of KDM2A peaks in KO B cells at TSS-proximal regions compared to WT B cells (Fig. 5c). Accordingly, H3K36me2 peak density was significantly reduced in KO B cells relative to WT B cells at these same regions and in the intragenic regions proximal to the TSS (Fig. 5c). Our findings indicate that the absence of miR-155 leads to an elevated expression of KDM2A, which in turn triggers enhanced binding of KDM2A to TSS-proximal regions and a concomitant depletion of H3K36me2 in those regions as well as in the intragenic regions.

To elucidate the biological functions of genes bound by KDM2A, we investigated the KDM2A target genes that displayed differential binding between WT and KO B cells and identified "WT-enriched" and "KO-enriched" KDM2A peaks (Supplementary Data 5). Functional annotation clustering analysis with DAVID[54] showed that genes with "WT-enriched" or "KO-enriched" KDM2A peaks were strongly associated with the regulation of mitochondrial functions (Fig. 5d). These results suggest that KDM2A plays a role in regulating the expression of nuclear mitochondrial genes during hypoxia. Among the genes with "WT-enriched" and "KO-enriched" KDM2A peaks, we identified 102 and 30 genes encoding nuclear mitochondrial proteins listed in MitoCarta 3.0[55], respectively. Bulk RNA-seq analysis showed that five mitochondrial genes with "KO-enriched" KDM2A peaks at promoter/TSS regions

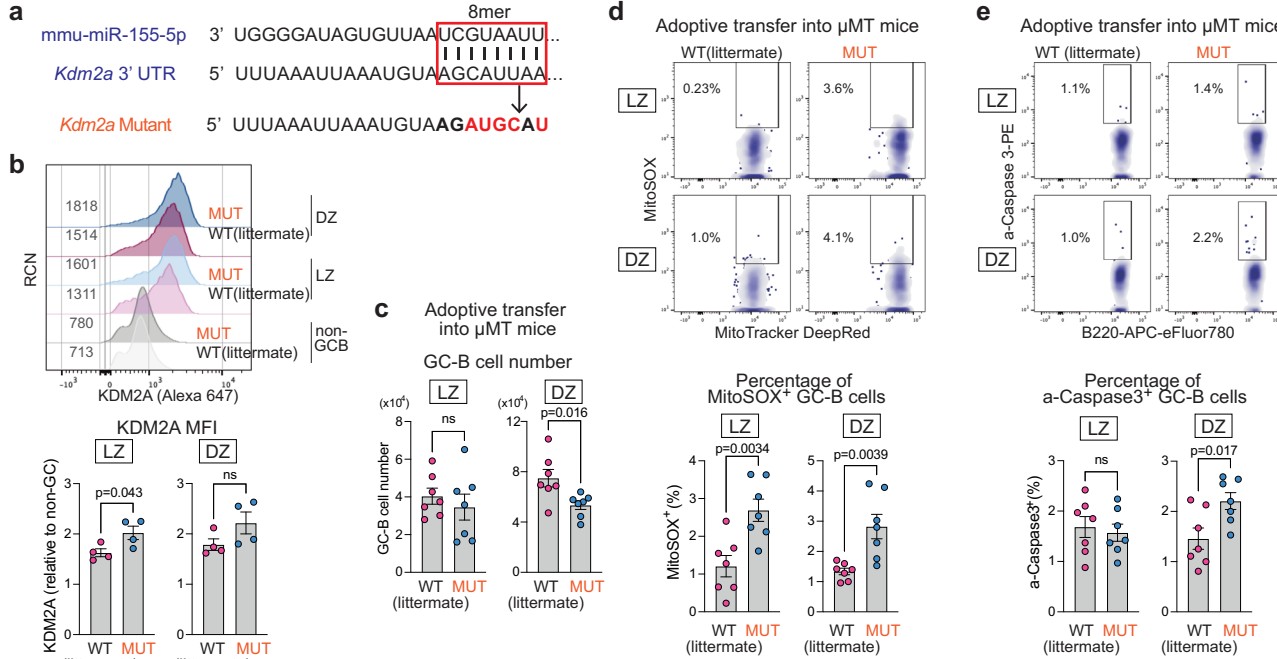

**Fig. 6 | The miR-155-*Kdm2a* interaction regulates mitochondrial ROS and apoptosis in GC-B cells. a** Schematics representation of the 8-mer miR-155-binding site in the 3'UTR of the *Kdm2a* (boxed) and the introduced base changes (red letters) to disrupt miR-155 binding in *Kdm2a* mutant mice. **b** Representative flow cytometric histograms showing KDM2A vs. RCN (top). RCN, relative cell number. Relative MFI values of KDM2A in LZ and DZ GC-B cells are shown (bottom). GC-B cells were derived from WT and MUT mice seven days after SRBC immunization. Unpaired student's t-test, two-tailed. Pooled from three independent experiments (WT *n* = 4; MUT *n* = 4). **c** Number of LZ and DZ GC-B cells from WT and MUT mice within the μMT adoptive transfer system. Unpaired student's t-test, two-tailed.

Pooled from two independent experiments (WT *n* = 7; MUT *n* = 7). **d** Representative flow cytometric plots of MitoTracker DeepRed vs. MitoSOX in GC-B cells from WT and MUT mice within the μMT adoptive transfer system (top). Percentage of MitoSOX$^+$ GC-B cells (bottom). Unpaired student's t-test, two-tailed. Pooled from two independent experiments (WT *n* = 7; MUT *n* = 7). **e** Representative flow cytometric plots of B220 vs. active-Caspase 3 in GC-B cells from WT and MUT mice within the μMT adoptive transfer system (top). Percentage of active-Caspase 3$^+$ GC-B cells (bottom). Unpaired student's t-test, two-tailed. Pooled from two independent experiments (WT *n* = 7; MUT *n* = 7). Unless otherwise stated, mean ± SEM is indicated. n.s., not significant.

and two mitochondrial ribosomal proteins with "KO-enriched" KDM2A peaks at the first intron, which corresponds to the intragenic regions proximal to the TSS, were reduced in their expression in KO cMyc$^+$ LZ GC-B cells compared to WT cMyc$^+$ LZ GC-B cells (Fig. 5e and Supplementary Data 2). The nuclear mitochondrial genes with "KO-enriched" KDM2A peaks exhibited a concomitant decrease in H3K36me2 levels (Supplementary Fig. 5a). These findings suggested that the differential binding of KDM2A between WT and KO B cells affected the expression of nuclear mitochondrial genes. Furthermore, among the five genes with "KO-enriched" KDM2A peaks located in the promoter/TSS regions, *Ndufv3* and *Atp5l* are known to play an essential role in OXPHOS[55]. Thus, the miR-155-*Kdm2a* axis has a critical function in regulating OXPHOS coupled with mitochondrial bioenergetic functions in activated B cells under hypoxic conditions.

## The miR-155-*Kdm2a* interaction regulates mitochondrial ROS and apoptosis in GC-B cells

To validate the regulatory interaction of the miR-155-*Kdm2a* axis, we generated mice with a mutated miR-155 binding site in the 3'UTR of *Kdm2a* (Fig. 6a). B cell development was normal in homozygous mutant (hereafter called "MUT") mice (Supplementary Fig. 6a). WT and MUT mice were immunized with SRBCs, and KDM2A expression in GC-B cells was assessed by flow cytometry. In LZ GC-B cells of MUT mice, KDM2A expression exhibited a ~1.2-fold increase relative to their WT littermates (Fig. 6b). No significant difference in KDM2A expression was observed in DZ GC-B cells. These results indicate that miR-155 limits *Kdm2a* expression in LZ GC-B cells through its binding site on *Kdm2a* mRNA. To examine B cell-intrinsic effects of the mutation, we performed adoptive transfer experiments by transferring B cells

isolated from the spleens of either WT or MUT mice into μMT mice, which lacks mature B cells[56]. The mice were then immunized with SRBCs. Seven days later, all recipient mice gave rise to GC-B cells, but the total number of DZ GC-B cells was reduced by ~29% in the MUT background compared to that of WT, while the number of LZ GC-B cells was similar between WT and MUT mice (Fig. 6c and Supplementary Fig. 6b, c).

We investigated the impact of the 3'UTR mutation on mitochondrial ROS and apoptosis in GC-B cells by transferring B cells into μMT mice. The proportion of MitoSOX$^+$ cells was higher in both LZ and DZ GC-B cells of MUT mice compared to WT (Fig. 6d). Moreover, the percentage of active-Caspase3$^+$ cells were increased in DZ GC-B cells of MUT mice, when compared to WT mice (Fig. 6e). The elevated apoptosis in GC-B cells from MUT mice compared to WT mice provides an explanation for the decreased number of DZ GC-B cells depicted in Fig. 6c. These results largely mirror the observations in *Mir155*-deficient mice. Moreover, the overexpression of KDM2A in GC-B cells through lentiviral transduction resulted in an increased percentage of MitoSOX$^+$ DZ GC-B cells compared to control cells (Supplementary Fig. 6d). Taken together, these findings underscore the crucial role of miR-155 in repressing *Kdm2a* to counteract the effects of hypoxia during the positive selection process.

## Discussion
We demonstrated that miR-155 critically regulates H3K36me2 levels by directly repressing histone demethylase *Kdm2a* during GC positive selection. This repression enabled GC-B cells to reprogram their metabolism through the remodeling of mitochondria and to initiate cell division in the hypoxic LZ microenvironment. Maintaining

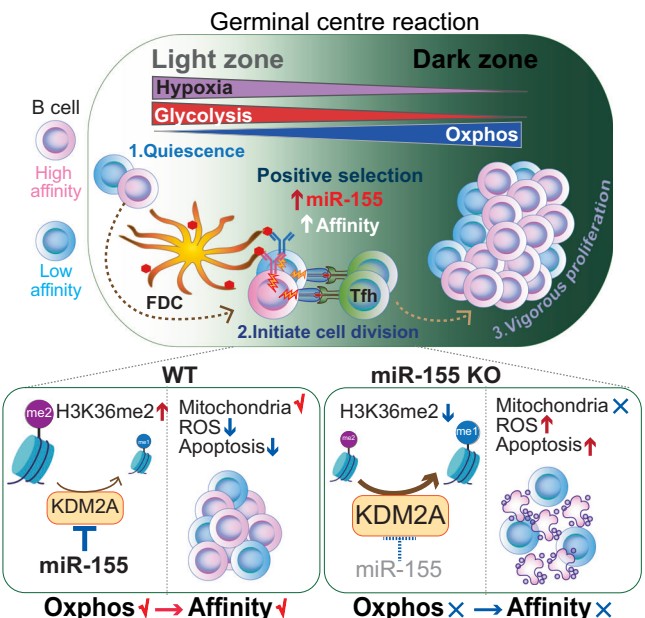

**Fig. 7 | Working model: miR-155-mediated epigenetic regulation control mitochondrial fitness during GC-B cell positive selection.** Following the reception of signals through BCR engagement and T cell help, LZ GC-B cells undergo a positive selection process that induces cell division. Positively selected GC-B cells, particularly high-affinity clones, migrate to the less hypoxic DZ and sustain their vigorous cell division by relying on OXPHOS for energy production. To promote the dynamic transition of GC-B cells from the hypoxic LZ to the DZ under hypoxia-induced stress, miR-155 plays a crucial role by repressing the hypoxia-induced expression of KDM2A. This repression mechanism enables precise regulation of H3K36me2 levels, leading to optimal expression of nuclear mitochondrial genes. As a result, mitochondrial remodeling is facilitated, and OXPHOS is enhanced in hypoxic conditions. Thus, the miR-155-*Kdm2a* interaction is indispensable for promoting mitochondrial fitness during positively selection, which provides robust mitochondrial in high-affinity clones to endure the high-rate of cell division in the DZ. This mechanism allows high-affinity clones to undergo clonal expansion, thus ensuring the process of affinity maturation.

mitochondrial fitness during positive selection in the LZ was essential for higher-affinity GC-B cells to undergo clonal expansion in the DZ, consequently ensuring affinity-maturation in GC responses. In the absence of miR-155, dysfunctional mitochondria resulted in increased ROS production and subsequent apoptosis in GC-B cells, which explains the compromised affinity maturation in miR-155-KO mice[22,57,58] (Fig. 7).

While *Mir155*-defcient mice show a compromised proliferation of GC-B cells, in vitro cultured KO B cells exhibit normal DNA synthesis and cell proliferation rates regardless of stimuli[57–59]. Thus, it was unclear why these in vitro cultured B cells cannot recapitulate the phenotype observed in vivo. In this study, we demonstrated that essential function of miR-155 is revealed under hypoxia in vitro and reproduced the known in vivo phenotypes observed in KO B cells. We have identified *Kdm2a* as a critical target gene of miR-155, which is induced under hypoxic conditions. *Kdm2a* exhibits context-dependent effects on proliferation[18,19] and may act as a negative regulator of proliferation in GC-B cells. Therefore, miR-155-mediated *Kdm2a* repression enabled GC-B cells to initiate cell division under hypoxic conditions in the LZ during positive selection.

Our findings demonstrated the critical role of miR-155 in facilitating the metabolic adaptation of positively selected GC-B cells for cell division while mitigating hypoxia-driven stress in the LZ. Our previous studies have shown that high-affinity clones display greater proliferative activity than low-affinity clones within cMyc⁺ LZ GC-B cell

subpopulation[7]. Indeed, miR-155 exerted its most potent suppressive effect on *Kdm2a* especially within the high-affinity clones of the cMyc⁺ LZ GC-B cell subpopulation (Fig. 3f). In agreement with this, *Mir155*-deficient mice exhibit a selective loss of high-affinity GC-B cells due to apoptosis[22,57,58]. In the absence of miR-155, cMyc⁺ LZ GC-B cells with higher levels of KDM2A experience detrimental effects such as increased ROS production due to dysfunctional mitochondrial and subsequent cell death (Fig. 1b, c, Fig. 6d, e). Despite the down-regulation of miR-155 expression in cMyc⁺ DZ GC-B cells and its absence in cMyc⁻ DZ GC-B cells, there was a persistent increase in ROS production in these cells when miR-155 was absent. This may be because DZ GC-B cells prioritize energy use for mitosis rather than mitochondrial repair. Therefore, GC-B cells appear to be incapable of restoring mitochondrial function after migration to the DZ. The findings underscore the importance of the miR-155-*Kdm2a* regulatory axis in the LZ, which promoted metabolic reprogramming and gradual enhancement of OXPHOS in positively selected GC-B cells. As a result, positively selected GC-B cells could cope with the higher rate of cell division in the DZ by utilizing ATP provided by OXPHOS in remodeled mitochondria. These results are consistent with a previous observation that GC-B cells with higher affinity require increased OXPHOS activity to undergo extensive proliferation[9].

KDM2A expression was upregulated >1.5-fold in *Mir155*-KO mice compared to WT mice in both cMyc⁺ and cMyc⁻ DZ GC-B cells (Fig. 3e). However, the expression level of KDM2A in DZ GC-B cells of MUT mice was comparable to that of WT mice. Firstly, there is evidence for a 6-mer miR-155 binding site in the 3'UTR of mouse *Kdm2a*[60]. Since this 6-mer binding site is intact in MUT mice, miR-155 may partially repress *Kdm2a* transcripts through this region, inhibiting the KDM2A elevation in MUT mice. Alternatively, other miR-155 target(s) could contribute to the induction of KDM2A. *Hif1a* can be directly targeted by miR-155[60] and potentially influences the expression of KDM2A because KDM2A may be induced partially in a HIF-1α-dependent manner under hypoxic conditions[41]. Given that *Hif1a* is repressed by miR-155 in a similar manner in both WT and MUT mice, the lower HIF-1α expression in MUT B cells compared to KO B cells may limit the efficiency of KDM2A induction in MUT mice.

MiRs generally have multiple targets. We identified 18 mRNAs that are predicted miR-155 targets in cMyc⁺ LZ GC-B cells and exhibit increased expression in the absence of miR-155 (Supplementary Fig. 3a). One of the molecules, *Jarid2* was previously discovered as a miR-155 target in positively selected GC-B cells, and its overexpression has been shown to increase apoptosis in LZ GC-B cells[22]. JARID2 functions as an accessory component of the Polycomb Repressive Complex 2 (PRC2), which catalyzes H3K27me3[61], thereby epigenetically regulating gene expression. Therefore, the modulation of PRC2 activity by JARID2 potentially contributes to elevated apoptosis in these cells. Another molecule, F-Box Only Protein 11 (*Fbxo11*), was also among the identified 18 genes, and is a part of a multi-subunit E3 ubiquitin-ligase protein complex that catalyze the ubiquitination of proteins destined for proteasomal degradation. GC-B cell-specific deletion of *Fbxo11* increases the number of GC-B cells, particularly DZ GC-B cells, suggesting that this molecule is associated with GC-B cells proliferation[62]. Therefore, the phenotypic alterations observed in *Mir155*-deficient mice may result from the cumulative effects of several causal targets acting together with *Kdm2a*.

Collectively, our findings demonstrate that miR-155 acts as an epigenetic regulator to limit the activity of *Kdm2a* on target genes, thereby maintaining appropriate levels of H3K36me2. The precise control of H3K36me2 levels during positive selection is vital for the optimal expression of nuclear mitochondrial genes in GC-B cells under hypoxic conditions, thus facilitating mitochondrial remodeling. The resultant metabolic reprogramming is indispensable for the survival and robust proliferation of GC-B cells within the DZ. Our findings provide insights that could pave the way for the development of

innovative therapeutics targeting histone PTMs to control GC responses in the context of vaccination and lymphomagenesis.

# Methods

## Mice

*Mir155*-deficient mice[59], SW_HEL mice[23], *Myc*gfp/gfp mice[24], *Aicda*-cre-hCD2[25], µMT (JAX:002288, The Jackson laboratories) and B6.SJL CD45.1 (strain code:494, Charles River Laboratories) mice were maintained on C57BL/6 J background. SW_HEL mice were interbred with *Myc*gfp/gfp and *Aicda*-cre-hCD2 mice and were maintained either *Mir155*-sufficient or *Mir155*-deficient background. B6.SJL CD45.1 mice were crossed with C57BL/6 J mice (JAX:000664, The Jackson laboratories), and heterozygous mice were used. In this study, male and female mice aged 6–25-weeks were used and euthanized by cervical dislocation. For the generation of *Kdm2a* mutant mice, a single guide RNA (AAAATTAAATGTAAGCATTAA) was designed using CRISPOR[63] to cut in the 3′UTR of the *Kdm2a* gene (Position 3369 – 3376 of *Kdm2a* 3′UTR, ENST00000398645.2). A 111 bp ssODN (CTTATTTAGAGTCGATCTCCAATGTTGTGCTAAGATTTTTAAATTAA ATGTAAGatgcAtGGGAATGAGTCTTAGGCCACCAGCTGACACATTGG CATTATTTTGGGTCAG) was designed containing the "mir155-2" mutation. Zygotes were generated by superovulation of C57BL/6 J mice and electroporated with 50 µL of Cas9 protein / sgRNA / ssODN (final concentrations 1.2 µM / 6 µM / 8 µM respectively). Both template and guide were added at the 1 cell stage. Electroporation of the embryos was performed using a Nepa21 electroporator (Sonidel). Two males (1 heterozygous and 1 homozygous) were found to have sequence perfect integration of the mir155-2 mutation via Sanger sequencing (Forward primer: CGAGGACGTAGAACATGCGA, Reverse primer: GGGCAGGCACTTCCTAGTTT) and were further bred to C57BL/6 J yielding 2 heterozygous F1s. Q-PCR CNV analysis further confirmed the presence of only one copy of the mir155-2 mutation. F1 mice were inbred to generate homozygous mutant mice. All mice were bred and kept under specific-pathogen-free conditions at the Francis Crick Institute Biological Research Facility in accordance with guidelines set by UK Home Office and the Francis Crick Institute Ethical Review Panel.

## Immunization, treatments, and adoptive transfers

Mice were immunized intraperitoneally with $2 \times 10^8$ SRBCs. For HEL-specific GC responses, splenic cells from donor mice were incubated with biotinylated antibodies: anti-CD43, anti-CD4 and anti-Ter-119, washed and incubated with anti-biotin microbeads (Miltenyi Biotec). B cells were enriched by biotin negative selection using MACS LS Columns (Miltenyi Biotec). The purity of recovered CD19+ B220+ B cells was > 97%. $5 \times 10^4$ to $1 \times 10^5$ HEL-specific B cells were transferred into B6.SJL CD45.1+ hosts intravenously. Mice were immunized intraperitoneally on the same day of cell transfer or one day later with $2 \times 10^8$ SRBCs conjugated with HEL[3×]. The HyHEL10 BCR of SW_HEL mice exhibits a binding affinity 13,000-fold lower to HEL[3×] than to HEL. Immunization with HEL[3×] induces robust and persistent GC responses at the expense of extrafollicular responses[64]. HEL[3×] protein was harvested from the supernatant of Chinese hamster ovary (CHO) cells expressing his-tagged HEL[3×][65] and was purified using a His-Trap FF column (Cytiva) on AKTA Pure column chromatography system (Cytiva). Eluted protein was concentrated to 0.5 ml (Vivaspin 15, 10,000 Da cutoff concentrator) and applied to a S200 10/300 analytical size exclusion column (Cytiva). HEL[3×] was found to eluted as a single, symmetrical peak and snap frozen at 1 mg/ml and stored at -80 °C until required. For B cell transfer from wild type (*Kdm2a*-mutant littermates) or *Kdm2a*-mutant mice, splenic B cells from donor mice were enriched by negative selection using MACS LS Columns as described above. $2.5 \times 10^7$ to $3 \times 10^7$ B cells were transferred into µMT hosts intravenously. Mice were immunized intraperitoneally on the same day of cell transfer with $2 \times 10^8$ SRBCs. To detect hypoxia, the

hypoxia dye MAR was used. MAR is non-fluorescent compound that is irreversibly converted into a fluorescent molecule by a reductase that is activated in response to low oxygen concentrations ( ~ 5% $O_2$)[44]. The conversion of MAR only occurs under hypoxic conditions[44], and the fluorescence of MAR is not affected by increased oxygen concentrations during tissue processing. Eight days after subcutaneous immunization of C57BL/6 mice with 50 µg of 4-Hydroxy-3-nitrophenyl acetyl hapten conjugated chicken gamma globulin (NP-CGG) in Imject Alum Adjuvant (ThermoFisher Scientific), 4.3 µg MAR hypoxia detecting probe (Goryo Chemical) in PBS per site or control DMSO in PBS was subcutaneously injected into two sites around the immunized inguinal lymph node. Three hours later, lymph nodes were harvested for flow cytometric analysis. Non-GC-B cells were gated as CD19+ B220+ CD38+ CD138- CD95- GL7-.

## Flow cytometry and cell sorting

Multicolor flow cytometry for analyses or for cell sorting was performed on LSR Fortessa or FACSAria (BD Biosciences) and data were analyzed using FlowJo v.10.8.1 software. PBS supplemented with 2% FBS was used as flow cytometry buffer. Dead cells were excluded using Zombie Fixable viability kit (BioLegend) during cell sorting. Harvested splenic B cells were enriched by CD43, Ter119 and CD4 biotin negative selection using MACS LS Columns to remove CD43+ plasmablasts and non-B cells for analysis. For HEL-specific GC responses, splenic B cells were enriched by CD45.1 negative selection using MACS LS Columns to remove recipient-derived cells. Single-cell suspensions were prepared and stained for the antibodies specific for cell surface markers. HEL-binding cells were stained with HEL (50 ng/ml), followed by HyHEL9 Alexa Fluor 647[64]. For active-caspase 3 staining, after surface marker staining, cells were fixed and permeabilized using a Fixation/Permeabilization Solution Kit (BD Biosciences) and stained with anti-active caspase 3 antibody (BD Biosciences). For KDM2A and H3K36me2 staining, cells were stained for surface marker antibodies and were fixed and permeabilized using a True-Nuclear Transcription Factor Buffer Set (BioLegend). The permeabilized cells were stained with anti-KDM2A or anti-H3K36me2antibody, followed by donkey anti-rabbit antibody staining. Cells stained with an anti-rabbit antibody, but no primary antibodies determined the basal level of fluorescence intensity for KDM2A/H3K36me2, and the basal fluorescence intensity was subtracted from each MFI of KDM2A/H3K36me2 staining for the calculation of relative MFI. For MitoSOX detection, cells were stained for surface marker antibodies. Then, they were incubated in HBSS with Calcium and Magnesium (ThermoFisher scientific) supplemented with 2.5 µM MitoSOX Red (ThermoFisher Scientific) and 50 nM Mito Tracker Deep Red FM (ThermoFisher Scientific) for 20 - 30 min at 37 °C and washed as described by the manufacturer before analysis.

## Antibodies

CD43, biotin (S7), BD Biosciences, 553269; CD4, biotin (GK1.5), Bio-Legend, 100404; Ter119, biotin (TER-119), BioLegend, 116204; CD45.1, biotin (A20), ThermoFisher Sciences, 13-0453-85; CD21/35, BV786 (7G6), BD Biosciences, 740894; CD23, FITC (B3B4), BD Biosciences, 553138; CD93, PE (AA4.1), ThermoFisher Sciences, 12-5892-82; CXCR4, PerCP-eFluor710 (2B11), ThermoFisher Sciences, 46-9991-82; CXCR4, APC (2B11), BD Biosciences, 558644; CD86, FITC (GL1), BD Biosciences, 553691; CD86, PECy7 (GL1), BioLegend, 105014; CD86, APCCy7 (GL1), BioLegend, 105030; CD86, BV605 (GL1), BioLegend, 105037; CD86, BV785 (GL1), BioLegend, 105043; CD95, PECy7 (Jo2), BD Biosciences, 557653; CD95, APC-R700 (Jo2), BD Biosciences, 565130; CD95, BV786 (Jo2), BD Biosciences, 740906; CD38, PerCP-eFluor710 (90), Thermo-Fisher Sciences, 46-0381-82; CD38, PECy7 (90), BioLegend, 102718; CD38, BV650 (90), BD Biosciences, 740489; CD38, BUV395 (90), BD Biosciences, 740245; GL7, Pacific blue (GL7), BioLegend, 144614; GL7, Alexa Fluor 488 (GL7), ThermoFisher Sciences, 53-5902-82; human CD2, PECy7 (TS1/8), BioLegend, 309214; human CD2, APC (TS1/8),

BioLegend, 309224; human CD2, Alexa700 (TS1/8), BioLegend, 309228; CD45.2, BV605 (104), BD Biosciences, 563051; CD45.2, BV786 (104), BD Biosciences, 563686; CD45.2, BUV395 (104), BD Biosciences, 564616; CD138, PE, (281-2), BioLegend, 142504; CD138, APC, (281-2), BioLegend, 142506; CD138, APCCy7, (281-2), BioLegend, 142530; CD138, BV605, (281-2), BioLegend, 142531; CD138, BV786, (281-2), BD Biosciences, 740880; B220, FITC (RA3-6B2), ThermoFisher Sciences, 11-0452-82; B220, APCCy7 (RA3-6B2), ThermoFisher Sciences, 47-0452-82; B220, BV421 (RA3-6B2), BioLegend, 103240; B220, BV650 (RA3-6B2), BioLegend, 103241; B220, BUV787 (RA3-6B2), BD Biosciences, 612838; CD19, FITC (6D5), BioLegend, 115506; CD19, BV605 (6D5), BioLegend, 115540; CD19, BV650 (6D5), BioLegend, 115541; CD19, PE-CF594 (1D3), BD Biosciences, 562291; IgM, BV605 (II/41), BD Biosciences, 743325; IgD, Alexa Fluor 647 (11-26 c.2a), BioLegend, 405708; IgG1, FITC (A85-1), BD Biosciences, 553443; IgG1, CF-594 (A85-1), BD Biosciences, 562559; Igκ light chain, BV421 (187.1), BD Biosciences, 562888; rabbit IgG, Alexa Fluor 647 (Poly4064), BioLegend, 406414; rabbit IgG, PE (Poly4064), BioLegend, 406421; rabbit IgG, Alexa Fluor 488 (Poly4064), BioLegend, 406416; CD40, purified (HM-40), BioLegend, 102902; active caspase 3, PE (C92-605), BD Biosciences, 550821; KDM2A, purified (EPR18602), Abcam, ab191387; cMYC, Alexa Fluor 555 (9E10), ThermoFisher Sciences, MA1-980-A555; Puromycin, Alexa Fluor 488 (12D10), Merck, MABE343-AF488; Histone H3K36me2, purified (EPR16994(2)), Abcam, ab176921; mouse IgM F(ab')$_2$ fragment, Jackson ImmunoResearch, 115-006-020; Streptavidin, PerCP-Cy5.5, BioLegend, 405214; Streptavidin, PE, ThermoFisher Sciences, 12-4317-87; Streptavidin, BV650, BioLegend, 405231; Streptavidin, BUV737, BD Biosciences, 612775.

## B cell culture, proliferation assays, ATP measurement and SCENITH

Harvested splenic B cells from *Mir155*-sufficient or *Mir155*-deficient C57BL/6 mice were enriched by CD43, Ter119 and CD4 biotin negative selection using MACS LS Columns. Purified B cells were cultured with positive selection mimicking stimulants (1 μg/ml anti-CD40 antibody (HM40-3), 5 μg/ml AffiniPure F(ab')$_2$ Fragment Goat Anti-Mouse IgM (H + L) (Jackson ImmunoResearch) and 4 ng/ml murine IL-4 (Pepro-Tech) in B cell medium (RPMI 1640 GlutaMAX Supplement medium supplemented with 10 % fetal bovine serum, 1 % pen-strep solution and 50 μM β-mercaptoethanol) at 37 °C in a humidified chamber containing 5% $CO_2$ and either 21% $O_2$ (normoxia) or 1% $O_2$ (hypoxia). For CTV proliferation assays, purified B cells were stained with 5μM CellTrace Violet (ThermoFisher Scientific) in PBS for 15 min at RT and washed as described by the manufacturer. For ATP measurement, purified B cells were cultured for 2 days as described above. On the measurement day, cells were washed twice in PBS to remove glucose and resuspended in either glucose medium (DMEM no glucose medium supplemented with 1 % pen-strep solution, 50 μM β-mercaptoethanol and 10 mM D-(+)-glucose) or galactose medium (DMEM no glucose medium supplemented with 1 % pen-strep solution, 50 μM β-mercaptoethanol and 10 mM galactose). After a thorough wash to remove glucose in the medium, $1 \times 10^5$ cells were plated per well and were further cultured in the presence or absence of 5 μM oligomycin A (Merck) at 37 °C in a humidified chamber containing 5% $CO_2$ and 1% $O_2$ for 2 h. All the cells were cultured in hypoxic conditions for 2 h to directly compare ATP production ability under the same condition. ATP was measured with Mitochondrial ToxGlo Assay kit (Promega) and followed the manufacturer's protocol. SCENITH was performed as previously described[33] with modifications. Splenic B cells derived from SW$_{HEL}$ adoptive transfer were first stained for surface marker antibodies, then divided into four groups and incubated for 30 min at 37 °C in B cell medium. Metabolic inhibitors (1 μM Oligomycin and/or 2-Deoxy-D-Glucose (2-DG) at the final concentration) or DMSO were then added, followed by a 15 min incubation at 37 °C. Puromycin (10 μg/ml final concentration) was directly added, and cells were incubated for an additional 15 min at

37 °C. Cells were washed and fixed using BD Cytofix/Cytoperm solution. Fixed cells were then stained intracellularly with anti-Puromycin antibody and washed immediately before flow cytometric analysis. All chemicals used for SCENITH were purchased from Merck. Glucose dependence and FAO/AAO (fatty acid oxidation and amino acid oxidation) capacity were calculated using the gMFI values of anti-Puromycin staining with the following formulas: Glucose dependence (%) = 100 × [(DMSO − 2DG) / (DMSO-2-DG & Oligomycin)], FAO/AAO capacity (%) = 100 − {100 × [(DMSO − 2DG) / (DMSO-2-DG & Oligomycin)]}.

## Immunofluorescence

Harvested spleens were embedded in OCT compound. 9 μm cryo-sections were fixed in ice cold acetone for 3 min, followed by air dry for 20 min. The sections were blocked with blocking buffer (0.3% Triton X-100, 1% BSA, 5% normal bovine serum and 1% rabbit serum in PBS) and stained with the described antibodies for overnight at 4 C°. For MitoTracker staining of cultured B cells, purified B cells were cultured for 2 days as described above and washed in PBS twice. Cell suspension containing $2 - 3 \times 10^6$ cells/ml in PBS was placed on Shi-fix coverslips (Shikhar Biotech) and incubated at 37 °C in a humidified chamber containing 5% $CO_2$ and 1% $O_2$ for 30–40 min. The coverslips were then washed as instructed by the manufacturer. Cells bound on the coverslips were stained with 83 nM MitoTracker Orange CM-H$_2$TMRos (ThermoFisher Scientific) and 50 nM Mito Tracker Deep Red FM at 37 °C for 15 min, followed by wash in warm PBS and fixation with 4% formaldehyde (Thermo-Fisher Scientific). Slides were mounted with Prolong Gold Antifade Reagent with DAPI (ThermoFisher Scientific), and images were acquired using LSM880 inverted confocal microscope (Zeiss) with 63 × objective and Zen Microscopy software (Zeiss).

## Image analysis

For the detection of mitochondrial ROS on confocal images of in vitro cultured B cells, the ratiometric cytoplasmic-to-nuclear intensity of mitochondrial ROS was quantified using a custom-built CellProfiler (v.4.2.5)[66] pipeline. Uneven illumination was first corrected in the Mito Tracker Deep Red FM and MitoTracker Orange CMTMRos channels by dividing the pixel values by a polynomial fit to the intensity of the image pixels. Following smoothing with median filtering, objects were then detected in the DAPI and Mito Tracker Deep Red FM channels using minimum cross-entropy (Mito Tracker Deep Red FM) or Otsu (DAPI) thresholding. All objects detected in the DAPI channel are then associated with corresponding objects in the Mito Tracker Deep Red FM channel, i.e., any Mito Tracker Deep Red FM objects without any corresponding DAPI objects are removed. The cytoplasm was then defined as the area within Mito Tracker Deep Red FM objects not occupied by DAPI objects. The mean intensity in the MitoTracker Orange CMTMRos channel was then measured within each cytoplasm object, and this was divided by the background value, the mean intensity in the MitoTracker Orange CMTMRos channel of each DAPI object to minimize batch effects. Cells with incorrect segmentation and apoptotic cells in the images were manually removed. For the analysis of co-expression of H3K36me2 / MYC and MYC / KDM2A in spleen sections was quantified using a custom FIJI script[67]. First, the approximate centers of cells were detected by identifying local extrema in eigenvalues of the Hessian tensor, calculated based on the IgD channel using FeatureJ (https://imagescience.org/meijering/software/featurej/hessian/). Cells were then fully segmented using these detected centers of cells as "markers" in a marker-controlled watershed operation, implementing using MorphoLibJ[68], using a Gaussian- then Sobel-filtered version of the IgD channel as input. The mean intensity of H3K36me2, MYC or KDM2A within the segmented cells was then quantified using ImageJ FIJI (v.2.9.0). All cells outside a manually defined region of interest were excluded.

## Analysis of histone PTMs

Cells were prepared for sorting by flow cytometry as described above. LZ or DZ GC-B cells ($>1 \times 10^5$) were directly sorted into 2 N $H_2SO_4$ as previously described[35]. After a 30 min incubation at room temperature, cellular debris was removed by centrifugation at $4000 \times g$ for 5 min and histones were precipitated from the supernatant with trichloroacetic acid at a final concentration of 20% (v/v) overnight at 4 °C. Precipitated histones were pelleted with centrifugation at $10,000 \times g$ for 5 min at 4 °C, washed once with 0.1% HCl in acetone, then once with 100% acetone with centrifugation at $15,000 \times g$ for 5 min at 4 °C. Histones were briefly dried with SpeedVac concentrator (Eppendorf) and stored at -80 °C until derivatization. Histones were propionylated and digested according to Garcia et al.[36] with the modification of a single round of propionylation for 1 hr prior to and following digestion. Targeted LC-MS/MS was performed on a Thermo TSQ Quantiva (ThermoFisher Scientific). All injections were performed in technical triplicate. Raw data were analyzed with Skyline (v.21.2)[69] according to published methods[35]. The relative abundance of each histone PTM was calculated from the total peak areas exported from Skyline. All the chemicals were purchased from ThermoFisher Sceintific.

## RNA-seq and data analysis

*Mir155*-sufficient or *Mir155*-deficient cMyc-GFP⁺ LZ GC-B cells were sorted by flow cytometry 5 days after $HEL^{3\times}$-SRBC immunization. Total RNA was prepared with TRIzol Reagent (ThermoFisher Scientific) combined with RNeasy Mini Kit (QIAGEN) and processed using a Tru-Seq Stranded mRNA Sample Prep kit (Illumina). RNA-sequencing was carried out on the Illumina HiSeq 2500 platform and generated ~43 million, 75 bp paired-end reads per sample. The RSEM package (v.1.2.31)[70] in conjunction with the bowtie2 alignment algorithm (v.2.2.9)[71] was used for the mapping and subsequent gene-level counting of the sequenced reads with respect to Ensembl mouse GRCm.38.86 version transcriptome. Normalization of raw count data and differential expression analysis was performed with the DESeq2 package (v.1.24.1)[72] within the R programming environment (v.3.6.0)[73]. Differentially expressed genes were defined as those showing statistically significant differences between KO and WT sample groups (FDR < 0.05). Majority of Mir155-deficient donor mice were males, whereas majority of Mir155-sufficient donor mice were female. Thus, the following 4 genes that are in the Y chromosome and more enriched in KO than WT mice were removed from the list of DEGs (ENSMUSG00000069045, ENSMUSG00000069049, ENSMUSG00000056673 and ENSMUSG00000068457). Gene list ranked by the Wald statistic were used to look for pathway and biological process enrichment using the Broad's GSEA software (v.2.1.0) with gene sets from MSigDB (v6)[74]. GSEA results were visualized with EnrichmentMap (v3.3.4)[75] in Cytoscape (v.3.7). We employed three miR target prediction tools, MiRDB[76], Diana[77] and Target scan[78] for identifying miR-155 targets.

## scRNA-seq and data analysis

Donor-specific *Mir155*-sufficient or *Mir155*-deficient GC-B cells were sorted 7 days after $HEL^{3\times}$-SRBC immunization. Libraries were prepared with Chromium Next GEM Single Cell 5′ Reagent Kit v2 (10 × Genomics) and sequenced on the Illumina NovaSeq 6000 platform. Fastq files were aligned and counted using CellRanger (v.6.1.2) and using as reference mm10-2020-A (obtained from 10 × Genomics). Matrices were imported into Seurat (v.4.0.5)[79], which was sued for downstream analysis. After QC, cells with less than 1000 features and more than 10% mitochondrial content were removed. Data was then normalized, scaled and variable features were identified per cell. Samples were integrated using "CCA" by integrating in the following order, first all replicates for each genotype and then by genotypes. DoubletFinder[80] was used to predict doublets. Clusters identified as contaminating for expression of Cd8a, Cd3e, Cd3d, Lyz2 and Gzma, and cells labelled as

doublets were removed from the analysis. Clusters were manually annotated using differential expressed genes obtained from FindMarkers. Uniform Manifold Approximation and Projection (UMAP) was used for visualization. VlnPlots and FeaturePlots have been used to display gene expression values per cluster or overlay on UMAP. The merged dataset of biological triplicates contained 25,358 and 24,087 GC-B cells for WT and KO cells, respectively. Unsupervised clustering using Seurat identified nine distinct clusters, each with their unique cell cycle status determined by the expression of genes associated with different phases of the cell cycle (Supplementary Fig. 1g). For gene signatures, a score was calculated using AddModuleScore prior to plotting. Statistical significance was obtained using Wilcoxon test and multiple comparison correction was performed using the "fdr" procedure. Differential gene expression between genotypes per cluster was carried out using a pseudobulk approach, in which gene counts were aggregated per sample and the "pseudoBulkDGE" function from the scran package[81] was used with the design -genotype. Genes are considered differentially expressed if FDR < 0.05. For trajectory analysis, Partition-based graph abstraction PAGA[29] was used. The integrated Seurat object was converted into a h5 format, neighbors were computed using 20 pcs and clusters detected using Seurat were used to compute the PAGA graph. Mitochondrial gene list was downloaded from MitoCarta3.0[55].

## ChIP-seq and data analysis

WT and KO B cells were cultured under hypoxic conditions with positive selection mimicking stimulants for 2 days. Cells were fixed for 1 hr in 2 mM EGS, followed by 15 min in 1% Formaldehyde. Chromatin was sheared using Bioruptor Pico sonication device (Diagenode). An aliquot of the chromatin was used to assess the size of the DNA fragments obtained by High Sensitivity NGS Fragment Analysis Kit (Agilent) on Fragment Analyzer (Agilent). ChIP was performed using IP-Star Compact Automated System (Diagenode). Chromatin corresponding to 13.5 µg and 1 µg were immunoprecipitated with KDM2A antibody (a gift from R. Klose, University of Oxford) and H3K36me2 (clone: EPR18602) (Abcam). Chromatin corresponding to 1% was used as Input. Libraries were prepared using MicroPlex Library Preparation Kit v3 (Diagenode) with 24 UDI for MicroPlex v3 (Diagenode). Generated libraries were purified using Agencourt AMPure XP (Beckman Coulter) and quantified using Qubit 1 × dsDNA HS Assay Kit (ThermoFisher Scientific). The fragment size was analyzed by High Sensitivity NGS Fragment Analysis Kit on Fragment Analyzer. Sequencing was carried out on the Illumina NovaSeq 6000 platform. Read alignment, library QC, peak detection and quantification was carried out using the nf-core/chipseq pipeline (v.1.2.2), using "-profile crick --min_reps_consensus 2". Briefly, reads were aligned using BWA[82] to GRCm38 using the genomes provided with the pipeline. Peaks were called using MACS2[83] and annotated using HOMER[84]. A list of consensus peaks is generated as the union of peaks detected in at least two replicates per condition. Bigwigs generated by the pipeline (scaled to 1 million mapped reads) were used to generate meta profile plots with Deeptools "compute matrix"[85]. A table containing CpG island containing genes was obtained from UCSC Table Browser[86]. Initially, an average of 26436 and 17586 KDM2A peaks were identified for WT and KO cells, respectively. The peaks that only one of the triplicate samples bound were removed from the subsequent analysis. After the exclusion of peaks, there were 9129 KDM2A peaks that were bound by both WT and KO B cells. The average fold change (fc) was calculated by dividing the average KDM2A peak signal by the input signal, and then "WT-enriched" or "KO-enriched" peaks among the 9129 KDM2A peaks bound by both WT and KO B cells were determined based on a fold change threshold (fc > 3) and a ratio of average fc between WT and KO cells ($> 2^{0.23}$-fold enrichment). Peaks with an fc > 3 from "peaks bound only by WT" (5905) and "peaks bound only by KO" (1479) were also categorized as "WT-enriched" and "KO-enriched" KDM2A peaks,

respectively. After applying the selection threshold, a total of 1403 genes with 1453 "WT-enriched KDM2A" peaks and 309 genes with 309 "KO-enriched KDM2A" peaks were identified (Supplementary Data 5). Among these, 73.6% and 78.0% of the "WT-enriched KDM2A" peaks and "KO-enriched KDM2A" peaks, respectively are genes containing CpG island.

## Statistical analysis

Statistical significance was determined by unpaired, two-tailed Student's t-test, Kruskal-Wallis one-way analysis of variance (ANOVA) with multiple comparisons, Wilcoxon rank sum test, or Kolmogorov-Smirnov test. The statistical method used is indicated in the respective Figure legends. Prism (v.9.4.1) software (GraphPad) and R were used to calculate all statistical analyses.

## Reporting summary

Further information on research design is available in the Nature Portfolio Reporting Summary linked to this article.

## Data availability

The scRNA-seq, bulk RNA-seq and ChIP-seq data generated in this study have been deposited in the NCBU Gene Expression Omnibus (GEO) database under accession number GSE230528, GSE231649, and GSE230527, respectively. Source data are provided with this paper and can be accessed at the following links: https://www.ncbi.nlm.nih.gov/geo/query/acc.cgi?acc=GSE230528, https://www.ncbi.nlm.nih.gov/geo/query/acc.cgi?acc=GSE231649, https://www.ncbi.nlm.nih.gov/geo/query/acc.cgi?acc=GSE230527. Source data are provided with this paper.

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

## Acknowledgements

This research was supported by the Francis Crick Institute core funding from Cancer Research UK, the UK Medical Research Council, and Wellcome Trust grant CC2078 (D.P.C.), BBSRC grant BBS/E/B/000C0407 (M.T.), and the National Institute of General Medical Sciences of the National Institutes of Health grant P41 GM108569 (N.L.K.). We thank M. Busslinger (Research Institute of Molecular Pathology) for *Aicda*-cre-hCD2 mice; B. P. Sleckman (University of Alabama at Birmingham) for *Myc*[gfp/gfp] mice; R. Brink (Garvan Institute of Medical Research/University of New South Wales) for SW[HEL] mice and HyHEL9 antibody; O. Bannard (University of Oxford) for providing CHO-his-tagged HEL[3×] cells; the staff of Genomics Facility of the Babraham Institute for technical support; the staff of the Francis Crick Institute Animal Facility, the Flow Cytometry Facility, the Advanced Sequencing Facility, the Advanced Light Microscopy Facility, and the Experimental Histopathology Facility for technical support; and E. Vigorito and C. Vinuesa for critical review.

## Author contributions

R.N. conceived and designed the study. M.L. and P.C. performed bioinformatic analysis. S.V-B. and G.D. generated *Kdm2a* mutant mice. R.N., J.M.C., and R.G. performed experiments and analyzed data. D.B. generated computational image analysis pipelines. N.P.B. and R.J.K. provided resources. N.L.K. provided supervision. M.T. interpreted data and critically revised the manuscript. R.N. and D.P.C. wrote the manuscript. All authors reviewed and edited the written manuscript.

## Funding

## Competing interests

The authors declare no direct competing financial or nonfinancial interests.
