## [Transparent Peer Review file · Nature Communications]

Epi-microRNA mediated metabolic reprogramming counteracts hypoxia to preserve affinity maturation

Corresponding Author: Dr Rinako Nakagawa

Version 0:

Reviewer comments:

Reviewer #1

(Remarks to the Author)

In this manuscript Nakagawa et al demonstrate that the loss of microRNA 155 (miR-155) leads to reduced numbers of Myc-positive germinal center (GC) B cells. At the same time these cells accumulate more mitochondrial ROS and show reduced expression of genes associated with mitochondrial activity. Moreover, the authors show that miR-155 regulates Kdm2a expression and thereby demethylation of H3K36me2, particularly in the high affinity clones of the GC exposed to hypoxia. Interestingly, miR-155 seems to be needed to maintain proliferation under hypoxia. Lastly, the authors show that mutating the miR-155 binding site in Kdm2a regulates dark zone GC numbers. MiR-155 has been demonstrated before to play a role in the survival of GC B cells, however the connection between Kdm2a expression and MiR-155 in GC B cells has not been explored before to my knowledge. However, although the authors discuss the role of miR-155 in "metabolic adaptation", this aspect is rather poorly supported by experimental data. Mitochondrial function has not been directly assessed experimentally. The performed experiments to measure ROS production lack important controls.

To strengthen the study the authors should in my opinion address the following concerns:

- The authors suggest that miR-155 leads to increased mitochondrial ROS production. However, MitoSOX, the dye used to measure mROS also stains dead cells. The authors should provide example experiments to show how they gated out dead cells and also include examples of the MitoSOX stain itself.
- On page 9 the authors analyse the gene expression profile of GC B cells in the presence or absence of miR-155 but make statements about "mitochondrial function". Gene expression is not always a good predictor of mitochondrial function. The authors should assess mitochondrial function experimentally. Since Seahorse metabolic flux analysis requires large cell numbers, the authors could utilize experimental methods more suitable for low-cell number analysis such as SCENITH.
- The dye used to stain hypoxic cells, MAR, seems to strongly stain a large part of non-GC B cells. This goes against observations made by other groups (e.g. Cho et al., 2016). The specificity of the dye should be demonstrated experimentally by for example staining B cells obtained from the blood and exposed to normoxia and hypoxia in vitro.
- The authors should provide information on cell survival under hypoxia in their performed in vitro experiments
- Hypoxia is usually associated with increased ROS production, however the authors observe less mROS in hypoxic cultures (Fig.4e). The authors should show cell size to ensure that reduced staining is not an artefact of changed cell size. Moreover, the authors should use antioxidants as controls for this staining to demonstrate specificity.
- The authors argue that miR-155 supports cell survival by regulating mitochondrial fitness and ROS production. However, it is possible that the observed increase in ROS production is simply a side-product of cell death and not a causative factor. The authors should treat the cells with anti-oxidants to test whether an increase in ROS production in the absence of miR-155 contributes to apoptosis induction.

Reviewer #2

(Remarks to the Author)

Germinal center B cells have complex metabolic requirements and intense and constantly shifting energy needs. T cell help

induction of MYC plays a central role in this process by enabling cell growth in positively selected cells. Expression of MYC can be toxic to cells and a previous report indicated that miR155 expressed simultaneously with MYC protects B cells from cell death. Based on these findings the authors hypothesized that miR155 does this by limiting mitochondrial ROS. Their excellent paper reveals an interesting new mechanism that resolves open questions in the immunology field regarding the function of miR-155 and regulation of mitochondrial energetics during affinity maturation in germinal centers.

Specifically, the authors show that miR-155 deletion increased ROS and apoptosis in germinal center B cells, and reduction in OxPhos related gene expression, including in MYC+ cells. They also show altered histone demethylation signatures, reduction in H3K36me2 by LC-MS, and corresponding increase in KDM2A. They connect these findings by demonstrating that KDM2A is a miR-155 target. Given the hypoxic germinal center environment and its relevance for metabolic programming they explore this aspect and show that miR-155 deletion results in higher ROS under more physiological hypoxic conditions, and observe increase ROS and reduced ATP production. Attempting to link this back to KDM2A they performed ChIP-seq experiments and show changes at gene promoters, and that such binding sites including mitochondrial pathways. Some of these had reduction in H3K36me2 and expression. The link between these observations was finally greatly strengthened by their generating a mouse model with mutated KDM6A miR-155 binding site, where they show similar phenotypes as miR-155 deletion. There is also a focused gain of function experiment where KDM6A induced ROS in B cells.

Overall, I thought this was an excellent paper, that addresses an important question and reveals interesting new information that will be viewed as impactful in the immunology field. I do have some suggestions to enhance impact and clarity, that could help the authors to enhance rigor and impact.

Figure 1:

Figure S1g does not sufficiently clearly indicate which cells in the UMAP express each signature. The authors need to find a more convincing way to document these findings, and indicate the proportion of DZ and LZ cells in their data.

The authors claim that their clusters represent a certain cell cycle order (s1e) but do not provide solid evidence to support this. I suggest to either remove this claim or perform more convincing analyses.

The PAGA analysis is interesting but a bit confusing when referring to LZ or DZ cells density of interaction. What is the message the authors wish to convey here? A clarification of this would be sufficient.

Figure 2.

What kinds of genes are reflected in the demethylase signatures? Are they actual demethylases in general, or is there an epigenetic theme? Although tables are provided should be further explained in the text.

Figure 2e: Are the levels of H3K36me2 also reduced in other cell populations? How specific are these effects? I would suggest the authors use flow cytometry to assess the relationship between K36me2 and its abundance among DZ, LZ and MYC positive cells in the miR-155 WT and KO setting.

Figure 5:

Figure 5a: What was the difference in peak numbers between miR-155 conditions?

What fraction of the promoter K36 and KDM2A peaks that are changing, overlap between conditions. I suggest to use correlation plots where can see if K36me2 peaks changing are the same KDM2A peaks, and if are positive or negative, or no correlation.

Figure 5b does not convincingly show a difference between genotypes, Figure 5c is better, but are the observed differences statistically significant?

Figure 6:

Was there increased apoptosis with overexpression of KMD6A? Perhaps this could help complete the idea and phenotype recapitulation.

Version 1:

Reviewer comments:

Reviewer #1

(Remarks to the Author)

The authors have included new experiments, controls and clarified statements and have in general answered all of my questions. I just have a few small suggestions:

- I am unable to find the description of the SCENITH experiments in the material and method section. Please add the

description of this method to the material and method section.

- Please add information on how often the experiment was repeated to the legend for supplementary figure 1E

- Please add information on how cells were stimulated to the legend for supplementary figure 4

-In response to my question about the dye MAR, the authors say that : "MAR detects ~5% O₂ (PMID: 24127124) whereas conventional hypoxia probes, such as pimonidazole, detect only conditions where O₂ is below 1% (PMID: 26780953)". This is important information that should be added to the main text (perhaps at p 16 line 17), since based on previous papers using pimonidazole staining, one would not expect non GC B cells to be hypoxic.

Reviewer #3

(Remarks to the Author)

The authors have satisfactorily addressed reviewers' comments.

Version 2:

Reviewer comments:

Reviewer #1

(Remarks to the Author)

The authors have addressed all of my remaining concerns/suggestions.

We thank the reviewers for their constructive comments and their interest in our manuscript. We have addressed their concerns individually as follows. The corresponding changes to the text have been made, which are highlighted in yellow, and have specified with the relevant page and line numbers.

Reviewer #1

1) The authors suggest that miR-155 leads to increased mitochondrial ROS production. However, MitoSOX, the dye used to measure mROS also stains dead cells. The authors should provide example experiments to show how they gated out dead cells and also include examples of the MitoSOX stain itself.

- We stained cells with Mito Tracker Deep Red which becomes positive only in live cells. To analyze ROS-positive cells, we gated cells that were positive for Mito Tracker Deep Red and MitoSox following gating for each individual GC subpopulation. We have included representative MitoSox/Mito Tracker Deep Red plots for each GC subpopulation as **Supplementary Fig. 1e** (pg7, ln15).

2) On page 9 the authors analyse the gene expression profile of GC B cells in the presence or absence of miR-155 but make statements about “mitochondrial function”. Gene expression is not always a good predictor of mitochondrial function. The authors should assess mitochondrial function experimentally. Since Seahorse metabolic flux analysis requires large cell numbers, the authors could utilize experimental methods more suitable for low-cell number analysis such as SCENITH.

- We carried out SCENITH analysis using GC-B cells derived from the SW_{HEL} adoptive transfer system. The results showed that miR-155 KO cMyc⁺ GC-B cells exhibited increased glucose dependence (i.e., reliance on Glycolysis and OXPHOS) and decreased fatty acid and amino acid oxidation (FAO/AAO) capacity compared to WT cMyc⁺ GC-B cells. These SCENITH results are consistent with observations in GC-B cells deficient in transcription factor A, mitochondrial (TFAM), which have defective mitochondria (please see Fig.6h of PMID: 37095377). This strongly indicates that miR-155 cMyc⁺ GC-B cells have defective mitochondrial functions. The results are included in the **Supplementary Fig. 1n** and the main text was changed accordingly (pg10, ln13-20).

3) The dye used to stain hypoxic cells, MAR, seems to strongly stain a large part of non-GC B cells. This goes against observations made by other groups (e.g. Cho et al., 2016). The specificity of the dye should be demonstrated experimentally by for example staining B cells obtained from the blood and exposed to normoxia and hypoxia in vitro.

- We thank the reviewer for noticing this discrepancy. MAR detects ~5% O₂ (PMID: 24127124) whereas conventional hypoxia probes, such as pimonidazole, detect only conditions where O₂ is below 1% (PMID: 26780953). Therefore, MAR can detect a wider range of hypoxia than conventional hypoxia probes. To experimentally

demonstrate the specificity of MAR, we stained B cells cultured for 2 days under either normoxic or hypoxic conditions. B cells cultured under hypoxia showed clearly higher fluorescence of MAR than those under normoxic culture. The results are included as **Supplementary Fig.4a** (pg16, Ln13-14).

4) The authors should provide information on cell survival under hypoxia in their performed *in vitro* experiments.

- In B cells cultured under hypoxic conditions, we observed significantly increased apoptosis compared to those cultured under normoxic conditions. In contrast to *ex vivo* GC-B cells, the levels of apoptosis between WT and KO B cells were similar *in vitro*. The results of the apoptosis assay, with active-Caspase 3 staining in B cells cultured for 2 days under either normoxic or hypoxic conditions, are included as **Supplementary Fig.4c** (pg17, Ln23 – pg18, Ln1).

5) Hypoxia is usually associated with increased ROS production, however the authors observe less mROS in hypoxic cultures (Fig.4e). The authors should show cell size to ensure that reduced staining is not an artefact of changed cell size. Moreover, the authors should use antioxidants as controls for this staining to demonstrate specificity.

- We cultured WT and KO B cells under normoxic and hypoxic conditions and compared their size (FCS) (**Supplementary Fig. 4d**). Both WT and KO B cells showed equally reduced cell size under hypoxic conditions compared to normoxic conditions. We agree with the reviewer that cell size may influence the levels of ROS MFI when comparing B cells cultured under normoxic conditions to those under hypoxic conditions. Consequently, we may have underestimated ROS MFI in KO B cells cultured under hypoxic conditions compared to those under normoxic conditions, despite the significant increase in ROS MFI under hypoxic conditions. We have added these considerations to the main text (pg18, Ln10-15).
- We carried out MitoSOX-based flow cytometric assay using B cells cultured for 2 days and confirmed the results obtained by imaging. These new results are included in **Supplementary Fig. 4e**. Additionally, we pre-treated the cells with the mitochondria-specific antioxidant, MitoTEMPO for 2.5 hours and tested ROS levels. This treatment reduced ROS levels in both WT and KO hypoxic cultures (Relative MitoSOX MFI to control cells of each condition is shown. Each dot represents one mouse). Therefore, we concluded that ROS production is specific to mitochondria in these cells (pg18, Ln17).

6) The authors argue that miR-155 supports cell survival by regulating mitochondrial fitness and ROS production. However, it is possible that the observed increase in ROS production is simply a side-product of cell death and not a causative factor. The authors should treat the cells with anti-oxidants to test whether an increase in ROS production in the absence of miR-155 contributes to apoptosis induction.

- As shown in the new **Supplementary Fig. 4c**, apoptosis levels did not differ between WT and KO cells after 2 days of hypoxic culture. This contrasts with *ex vivo* GC-B cells, where clear differences were observed between WT and KO, as shown in **Fig. 1c**. In contrast, the impact of miR-155 deficiency on proliferation in *ex vivo* GC-B cells (PMID: 26657861 and this manuscript), potentially caused by mitochondrial defects, is well recapitulated in the *in vitro* system (Fig. 4c). The previous version of our manuscript lacked confirmatory data showing that miR-155 deficiency impacts proliferation in *ex vivo* GC-B cells. We have now included **Supplementary Figure 1f**, which confirms that proliferation, assessed by EdU incorporation, is compromised in KO GC-B cell subsets compared corresponding WT GC-B cell subsets (pg7, ln16,17).

Reviewer #2

Figure 1:

1) Figure S1g does not sufficiently clearly indicate which cells in the UMAP express each signature. The authors need to find a more convincing way to document these findings, and indicate the proportion of DZ and LZ cells in their data.

- To improve data visualization, we have replaced the UMAP plots with dot plots showing the proportion of DZ and LZ cells in each cluster. The new plots are shown in **Supplementary Fig. 1i**.

2) The authors claim that their clusters represent a certain cell cycle order (s1e) but do not provide solid evidence to support this. I suggest to either remove this claim or perform more convincing analyses.

- We analyzed temporal dynamics of DZ clusters, 3, 4, 6 and 7 using RNA velocity. The analysis shows the temporal sequence of DZ clusters as Cl6→ 4 → 7 → 3. The new plot is included in **Supplementary Fig. 1g** (pg8, ln12-13,14).

3) The PAGA analysis is interesting but a bit confusing when referring to LZ or DZ cells density of interaction. What is the message the authors wish to convey here? A clarification of this would be sufficient.

- We employed PAGA for trajectory analysis in our data, following its capability for analyzing datasets where circular trajectories are expected (PMID: 30936559).

PAGA generates a graph-like topology by estimating connectivity between partitions. It groups cells in partitions (nodes in the graph) based on transcription similarity and predicts connectivity between these partitions (edges in the PAGA graph, representing neighborhood relations). PAGA aligns cells to trace a putative biological process from progenitor cells to different fates, following paths along nodes in the graph, and averages these paths to outline putative biological process.

- To establish connections between partitions, PAGA builds a statistical model that compares the paths from cells between partitions. If the number of similarities between cells from two given partitions exceeds similarities found by random chance, an edge is drawn between them. The width of edges in the PAGA graph reflects the strength of connectivity, indicating the presence of a genuine connection. In our data, the thickest edges in the PAGA graph represent the connection between cluster 0 and cluster 3, corresponding to the transition between DZ-GC-B cells and LZ-GC-B cells, and between cluster 1 and cluster 8, corresponding to the transition between LZ-GC-B cells and pre-memory B cells. To aid in understanding the connectivity between clusters, we included an “edge weight indicator” in **Fig.1d**.

Figure 2:

4) What kinds of genes are reflected in the demethylase signatures? Are they actual demethylases in general, or is there an epigenetic theme? Although tables are provided should be further explained in the text.

- We have included the lists of genes within the four gene sets (GO demethylation, GO Histone demethylase activity, GO protein demethylation and GO demethylase activity) that appeared as enriched signatures in KO cMyc⁺ LZ GC-B cells compared to WT cMyc⁺ LZ GC-B cells, as **Supplementary Table 3**. The “GO demethylation” gene set comprises general demethylases, whereas “GO histone demethylase activity” includes epigenetic demethylases (pg11, ln8-9, 11).

5) Figure 2e: Are the levels of H3K36me2 also reduced in other cell populations? How specific are these effects? I would suggest the authors use flow cytometry to assess the relationship between K36me2 and its abundance among DZ, LZ and MYC positive cells in the miR-155 WT and KO setting.

- We assessed H3K36me2 levels by flow cytometry and confirmed that H3K36me2 is decreased in KO cMyc⁺ LZ GC-B cells compared to WT counterparts. Moreover, consistent with the MS analysis, H3K36me2 levels were reduced in total LZ GC-B cells derived from KO mice compared to WT, but not in total DZ GC-B cells. These results are included in **Fig. 2e, Supplementary Fig. 2d** (pg13, ln16-20).

Figure 5:

6) Figure 5a: What was the difference in peak numbers between miR-155 conditions? What fraction of the promoter K36 and KDM2A peaks that are changing, overlap between conditions. I suggest to use correlation plots where can see if K36me2 peaks changing are the same KDM2A peaks, and if are positive or negative, or no correlation. Figure 5b does not convincingly show a difference between genotypes, Figure 5c is better, but are the observed differences statistically significant?

- The average peak number with triplicate samples for KDM2A is 26436 in WT B cells and 17586 in KO B cells. The average peak number with triplicate samples for H3K36me2 is 89,764 in WT B cells and 63,116 in KO B cells.
- To generate correlation plots, we needed to identify H3K36me2 peaks in the exact same regions where KDM2A peaks are located within the same type of cells. This was impractical, as the reduction of H3K36me2 sometimes resulted in no detectable peaks in these regions. Therefore, to determine the levels of H3K36me2 in the TSS/promoter regions of genes where both WT and KO cells have KDM2A peaks, we took a different approach. First, we created bed files for CpG-containing genes for the first 2.5 kb downstream of a TSS and counted the occurrence of H3K36me2 peaks from WT and KO using “bedtools coverage – a bed bam” (PMID: 20110278). We then normalized the coverage data with size factors obtained by dividing the number of mapped reads (extracted using “samtools flagstat”) by 10 million reads (PMID: 19505943), and created a ratio of average KO/WT. The results are included as **Supplementary Table 4**. Compared to WT cells, we found that KO cells exhibit higher H3K36me2 levels in the TSS regions of 6,724 genes, while KO cells exhibit lower H3K36me2 levels in the TSS regions of 17,618 genes. Therefore, we confirm that H3K36me2 levels are more reduced in KO cells than in WT cells at peaks in the TSS/promoter regions of CpG containing genes (pg20, ln15-19).
- To demonstrate statistical differences in peak distribution, we used “Kolmogorov-Smirnov test”. The test results showed that the distribution of both KDM2A and H3K36me2 peak is statistically significant between WT and KO cells. We have added the description of statistics in the figure legend and results in **Fig. 5c** (pg20, ln23, pg21, ln2).

Figure 6:

7) Was there increased apoptosis with overexpression of KMD6A? Perhaps this could help complete the idea and phenotype recapitulation.

- We assume the reviewer means overexpression of KDM2A, although it's written as “KMD6A”. Similarly to the observations in miR-155 KO mice, we observed significantly increased apoptosis in GC-B cells in *Kdm2a* mutant mice (**Fig.6e**). We also examined GC-B cells with overexpressed KDM2A and found that these cells had increased ROS levels (**Supplementary Fig. 6d**), consistent with the findings in *Kdm2a* mutant mouse experiments (**Fig.6d**).

We thank the reviewers for their time and interest in our manuscript. We have addressed their concerns as follows. The corresponding changes have been made to the texts, which are highlighted in yellow, with the relevant page and line numbers.

Reviewer #1

1) The authors have included new experiments, controls and clarified statements and have in general answered all of my questions. I just have a few small suggestions:

- I am unable to find the description of the SCENITH experiments in the material and method section. Please add the description of this method to the material and method section.

The SCENITH protocol has been added to the Material and Method section (pg 32, line 22-23, pg34, line 3-16).

2)- Please add information on how often the experiment was repeated to the legend for supplementary figure 1E

We have added the number of times the experiments were repeated in Extended Figure 1E figure legend (Supplementary materials, pg2, line 6).

3)- Please add information on how cells were stimulated to the legend for supplementary figure 4

We have added the B cell stimulants used for all experiments in Extended Figure 4 (Supplementary materials, pg6, line 15-16).

4) -In response to my question about the dye MAR, the authors say that : "MAR detects ~5% O₂ (PMID: 24127124) whereas conventional hypoxia probes, such as pimonidazole, detect only conditions where O₂ is below 1% (PMID: 26780953)". This is important information that should be added to the main text (perhaps at p 16 line 17), since based on previous papers using pimonidazole staining, one would not expect non GC B cells to be hypoxic.

We have added a new sentence explaining that the difference in detection sensitivity between MAR and conventional hypoxia probes, such as pimonidazole, may contribute to the detection of non-GC-B cells as hypoxic (pg 16, line 17-20).

We sincerely thank all the reviewers for their valuable time and insightful feedback on our manuscript. We believe that we have addressed all their concerns and concerns.

REVIEWERS' COMMENTS

Reviewer #1 (Remarks to the Author):

The authors have addressed all of my remaining concerns/suggestions.